# An aberrant phase transition of stress granules triggered by misfolded protein and prevented by chaperone function

Daniel Mateju[1], Titus M Franzmann[1], Avinash Patel[1], Andrii Kopach[1], Edgar E Boczek[1], Shovamayee Maharana[1], Hyun O Lee[1], Serena Carra[2], Anthony A Hyman[1] & Simon Alberti[1,*]

## Abstract

Stress granules (SG) are membrane-less compartments involved in regulating mRNAs during stress. Aberrant forms of SGs have been implicated in age-related diseases, such as amyotrophic lateral sclerosis (ALS), but the molecular events triggering their formation are still unknown. Here, we find that misfolded proteins, such as ALS-linked variants of SOD1, specifically accumulate and aggregate within SGs in human cells. This decreases the dynamics of SGs, changes SG composition, and triggers an aberrant liquid-to-solid transition of *in vitro* reconstituted compartments. We show that chaperone recruitment prevents the formation of aberrant SGs and promotes SG disassembly when the stress subsides. Moreover, we identify a backup system for SG clearance, which involves transport of aberrant SGs to the aggresome and their degradation by autophagy. Thus, cells employ a system of SG quality control to prevent accumulation of misfolded proteins and maintain the dynamic state of SGs, which may have relevance for ALS and related diseases.

**Keywords** protein aggregation; protein misfolding; proteostasis; SOD1; stress granules
**Subject Categories** Membrane & Intracellular Transport; Protein Biosynthesis & Quality Control; RNA Biology
**The EMBO Journal (2017) 36: 1669–1687**

See also: **P Siwach & D Kaganovich** (June 2017)

## Introduction

RNP granules are membrane-less compartments formed by RNAs and RNA-binding proteins (RBPs). These compartments have diverse roles in RNA processing, transport, storage, or degradation (Darnell, 2013; Mitchell & Parker, 2014; Singh *et al*, 2015). One type of RNP compartment are stress granules (SGs), which

are involved in the cellular stress response and the regulation of mRNA translation. In stressed cells, SGs assemble to sequester non-translating mRNAs together with RBPs and other factors involved in translation repression (Anderson & Kedersha, 2008; Buchan & Parker, 2009). Recent findings suggest that SGs may form by a process known as liquid–liquid phase separation and exhibit rapid assembly kinetics and liquid-like properties such as fusion or fission (Hyman *et al*, 2014; Kroschwald *et al*, 2015; Molliex *et al*, 2015; Patel *et al*, 2015).

The rapid and transient assembly of SGs relies on particular RBPs that harbor self-interacting domains of low sequence complexity (LC domains; Gilks *et al*, 2004; Decker *et al*, 2007; Han *et al*, 2012; Kato *et al*, 2012; Molliex *et al*, 2015; Patel *et al*, 2015). It is now emerging that these RBPs also form pathological aggregates in neurons of patients afflicted with neurodegenerative diseases (Li *et al*, 2013; Ramaswami *et al*, 2013; Robberecht & Philips, 2013). For example, the SG component TDP-43 has been identified in pathological inclusions of Alzheimer's, Huntington's, Parkinson's, FTD (frontotemporal dementia), and ALS patients (Dewey *et al*, 2012). Additional SG components with LC domains, such as FUS, TIA-1, Ataxin-2, or hnRNPA1, have also been linked to protein aggregates in ALS patients (Vance *et al*, 2009; Elden *et al*, 2010; Kim *et al*, 2013a; Li *et al*, 2013; Ramaswami *et al*, 2013). These observations led to the hypothesis that pathological inclusions of RBPs are derived from SGs (Dewey *et al*, 2012; Wolozin, 2012; Bentmann *et al*, 2013; Li *et al*, 2013; Aulas & Vande Velde, 2015). However, direct evidence that physiological SGs can mature into pathological inclusions is still lacking.

Recent *in vitro* studies with RBPs involved in ALS support the hypothesis of a slow maturation of RNP granules into pathological aggregates. Purified SG components, such as FUS or hnRNPA1, have been shown to phase separate into liquid droplets *in vitro* (Molliex *et al*, 2015; Murakami *et al*, 2015; Patel *et al*, 2015). These liquid compartments convert with time into a more solid state that is reminiscent of pathological aggregates commonly seen in patients. Moreover, the conversion from a liquid to an aberrant solid-like state is accelerated by

1  Max Planck Institute of Molecular Cell Biology and Genetics, Dresden, Germany
2  Department of Biomedical, Metabolic and Neural Sciences, University of Modena and Reggio Emilia, Modena, Italy
    *Corresponding author. Tel: +49 351 210 2663; E-mail: alberti@mpi-cbg.de

ALS-linked mutations in FUS and hnRNPA1 (Molliex *et al*, 2015; Murakami *et al*, 2015; Patel *et al*, 2015). However, it still remains unclear whether similar aberrant phase transitions occur in cells. In contrast to the reconstituted compartments, SGs have a multicomponent composition. Furthermore, SGs exist in a complex cellular environment, where they are affected by additional factors such as chaperones and pathways for protein degradation. Indeed, ALS-associated mutations in FUS that promote a liquid-to-solid phase transition *in vitro* do not seem to have major effects on SG dynamics in cultured cells (Patel *et al*, 2015, and unpublished observations). This suggests that cells have an extraordinary capacity to maintain the liquid state of SGs, despite the strong drive of numerous RBPs to misfold and aggregate. This raises two important questions: How is the physiological liquid-like state of SGs maintained in living cells? And what triggers the conversion of SGs into pathological aggregates?

Neurodegenerative disorders such as ALS are late-onset, suggesting that the physiological decline associated with aging could contribute to the conversion of SGs into pathological inclusions. One of the hallmarks of aging cells is a progressive decline of the proteostasis machinery regulating the abundance of misfolded proteins (Taylor & Dillin, 2011). If misfolded proteins are left uncontrolled, they can acquire toxic properties and compromise cell viability. Therefore, cells devote a substantial amount of resources into protein quality control (PQC; Kim *et al*, 2013b). The PQC machinery consists of an extensive network of chaperones, co-chaperones, and protein degradation pathways, which are able to recognize misfolded proteins and facilitate their refolding or degradation. Interestingly, some cases of ALS have been linked to mutations in PQC components such as VCP/p97 or ubiquilin-2 (Deng *et al*, 2011; Buchan *et al*, 2013; Majcher *et al*, 2015). Moreover, about 10% of familial forms of ALS are associated with misfolding and aggregation of the cytosolic protein SOD1 (superoxide dismutase 1; Robberecht & Philips, 2013; Bunton-Stasyshyn *et al*, 2015). However, in contrast to misfolding-prone RBPs, such as TDP-43 and FUS, there is no obvious link between SOD1 and SGs. This has led to the proposal of two distinct protein aggregation pathways—one involving RBPs and one involving SOD1. However, previous studies have shown that proteins in heat shock granules, which share many components with human SG, extensively interact with misfolded proteins in species such as budding yeast and *Drosophila* (Cherkasov *et al*, 2013; Kroschwald *et al*, 2015). This suggests that a similar interplay between SGs, misfolded proteins, and PQC may also exist in human cells, which may be relevant for age-related diseases.

Here, we demonstrate that misfolded proteins, including ALS-associated variants of SOD1, have a tendency to accumulate and aggregate within SGs in human cells. The presence of misfolded proteins in SGs triggers a liquid-to-solid phase transition of SGs and promotes the recruitment of chaperones. We find that the chaperone HSP70 has a key role in preventing the formation of aberrant SGs and is required for SG disassembly. In addition, we show that aberrant SGs can be transported to the aggresome where they are degraded by autophagy. Thus, we propose that cells have evolved a complex system of SG surveillance that serves to maintain SG composition, integrity, and dynamics.

# Results

## SGs co-assemble with misfolded proteins including ALS-associated variants of SOD1

Recent studies showed that RBPs and RNAs can undergo liquid–liquid phase separation and form liquid compartments *in vitro* (Elbaum-Garfinkle *et al*, 2015; Lin *et al*, 2015; Molliex *et al*, 2015; Patel *et al*, 2015; Zhang *et al*, 2015). It has been proposed that this physical principle underlies the formation of liquid-like SGs in living cells. Because stressed and aging cells produce high levels of misfolded proteins, we hypothesized that these misfolded proteins could affect the dynamic properties of SGs. To test whether there is an interaction between SGs and misfolded proteins, we first purified the SG component FUS and the model misfolding-prone protein Ubc9TS, which contains a single amino acid substitution conferring a conformational instability (Betting & Seufert, 1996; Kaganovich *et al*, 2008). Fluorescence spectroscopy analysis confirmed that purified Ubc9TS has a less compact conformation and higher surface hydrophobicity than the control protein Ubc9WT, consistent with misfolding (Appendix Fig S1A and B). Next, we tested whether misfolded Ubc9TS has a tendency to interact with *in vitro* reconstituted FUS compartments. Interestingly, misfolded Ubc9TS accumulated in FUS compartments more strongly than Ubc9WT (Fig 1A and Appendix Fig S1C). This suggests that misfolded proteins may have a tendency to accumulate in phase-separated liquid compartments.

To detect SGs in living cells, we generated stable HeLa cell lines expressing fluorescently tagged SG components from bacterial artificial chromosomes (BACs; Fig EV1A and B; Poser *et al*, 2008). BACs carry the genomic DNA including the endogenous promoter and regulatory sequences, thereby allowing the transgene to be expressed at physiological levels (Fig EV1A). To study the interaction between SGs and misfolded proteins, we employed model misfolding-prone proteins such as Ubc9TS, VHL (Von Hippel–Lindau), and ALS-linked mutant SOD1. Previous findings showed that proteasome inhibition leads to the accumulation of these misfolded proteins in the aggresome, a large perinuclear inclusion targeted to degradation by autophagy (Corcoran *et al*, 2004; Weisberg *et al*, 2012; Yung *et al*, 2015). In agreement with this, Ubc9TS and SOD1(A4V) accumulated in aggresomes in cells treated with the proteasome inhibitor MG132 (Fig EV1C). Next, we investigated whether aggresomes and SGs interact. We observed that G3BP-labeled SGs assembled after ~3 h of MG132 treatment, but disappeared after ~6 h, shortly after aggresomes started to form (Fig EV1D and Movie EV1). No colocalization between SGs and aggresome precursors was observed, indicating that under these conditions, cells are able to keep these structures separate.

Next, we used heat stress to simultaneously induce protein misfolding and SG formation. Interestingly, temperature-induced SGs often colocalized with misfolded Ubc9TS, but not with Ubc9WT (Fig 1B), consistent with the *in vitro* results. Ubc9TS-positive SGs contained SG markers such as FUS (Fig 1B), G3BP (Fig 1C), or eIF3η (Appendix Fig S2A). However, not all cells showed this phenotype—in some cells, Ubc9TS remained diffusely distributed or aggregated in separate foci (Appendix Fig S2B).

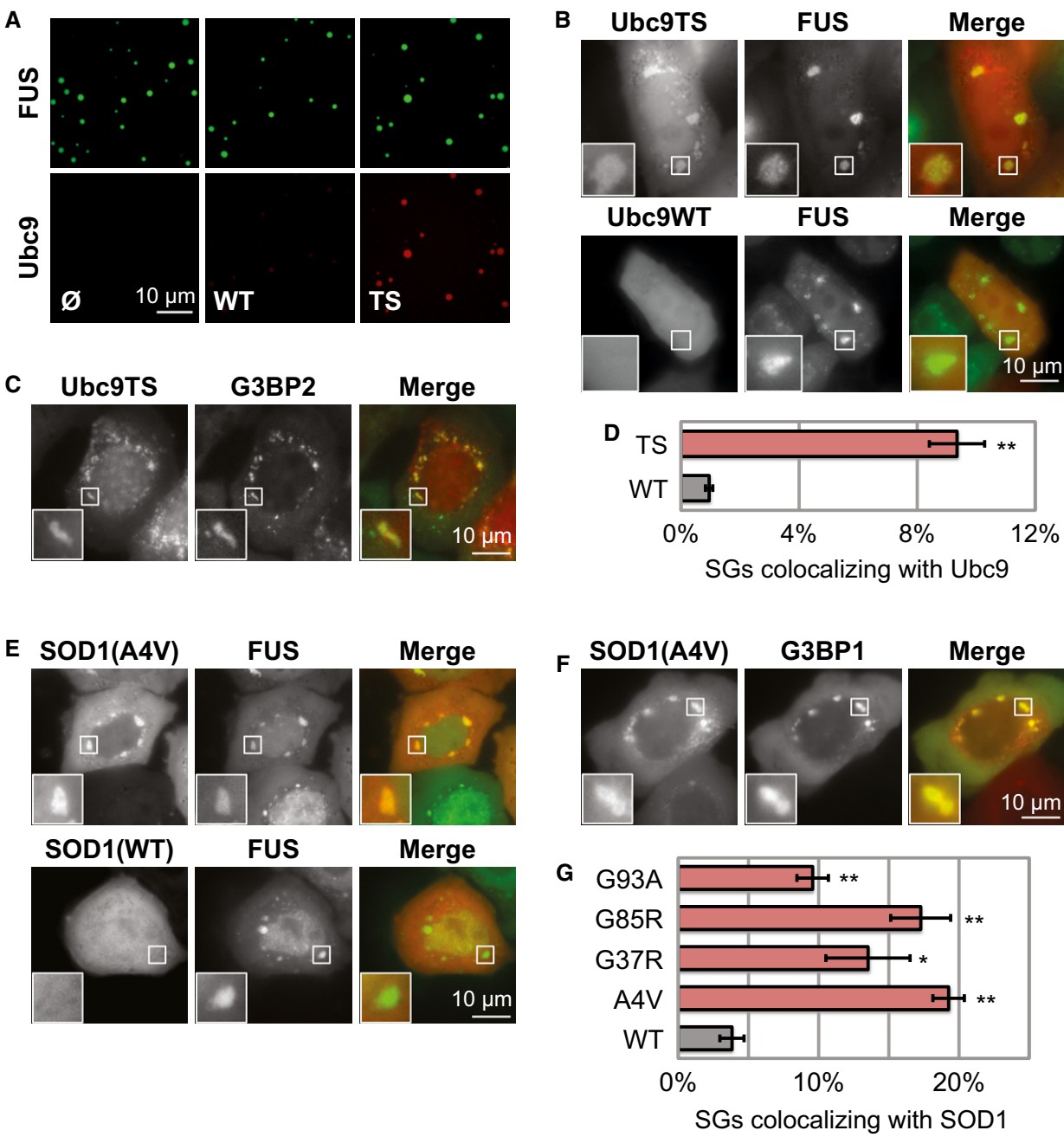

**Figure 1. SGs co-assemble with misfolded proteins including ALS-associated SOD1.**

A Purified Ubc9TS accumulates in liquid compartments formed by 5 μM FUS(G156E)-GFP *in vitro*. The Ubc9 concentration is at 4 μM.

B Localization of Ubc9TS-mCherry or Ubc9WT-mCherry co-expressed with FUS-GFP in live HeLa cells exposed to heat stress (2 h at 43°C).

C Ubc9TS-positive SGs contain G3BP2-GFP.

D Fraction of SGs colocalizing with Ubc9TS-mCherry or Ubc9WT-mCherry in cells expressing G3BP2-GFP and heat-stressed for 2 h. A total of 11,794 SGs were analyzed. Colocalization is defined by fluorescent ratio > 1.4. Mean values from three experiments are plotted. Error bars = SEM. **$P$ < 0.01 (*t*-test, compared to WT).

E Localization of SOD1(A4V)-GFP or SOD1(WT)-GFP co-expressed with FUS-mCherry in live cells exposed to heat stress (2 h at 43°C).

F SOD1-positive SGs contain G3BP1-mCherry.

G Fraction of SGs colocalizing with different variants of SOD1-GFP in cells expressing FUS-mCherry and heat-stressed for 2 h. A total of 12,028 SGs were analyzed. Error bars = SEM. *$P$ < 0.05, **$P$ < 0.01 (*t*-test, compared to WT).

Using a high-content automated imaging assay, we estimated that over 9% of SGs were highly enriched for Ubc9TS, while less than 1% were enriched for Ubc9WT (Fig 1D). We refer to SGs that do not accumulate misfolded proteins as "normal SGs" and those that accumulate misfolded proteins as "aberrant SGs". Similarly to misfolded Ubc9TS, we observed that misfolded SOD1(A4V)

localized to SGs induced by heat stress, while wild-type SOD1 remained diffusely distributed (Fig 1E). SOD1-positive SGs contained markers such as FUS (Fig 1E), G3BP (Fig 1F), or eIF3η (Appendix Fig S2A). As for Ubc9TS, some cells showed a different phenotype, with SOD1(A4V) remaining diffusely distributed or aggregating in separate foci (Appendix Fig S2B). Using a high-content automated imaging assay to compare the distribution of different SOD1 variants, we find that all the tested ALS-linked SOD1 variants have a tendency to accumulate in SGs compared to wild-type SOD1 (Fig 1G). As these SOD1 variants are prone to misfolding and aggregation (Rakhit *et al*, 2007; Prudencio *et al*, 2009), the results suggest that misfolded SOD1 is recruited to SGs in heat-stressed cells. We conclude that misfolded proteins have a strong tendency to accumulate in SGs and that the folding state determines whether a protein will interact with SGs or not. In further experiments, we use heat stress as an experimental condition, unless otherwise stated.

## Misfolded proteins form aggregates after accumulating in SGs

We have shown that misfolded SOD1 accumulates in SGs during heat stress. Misfolded SOD1 is known to form stable inclusions (Chattopadhyay & Valentine, 2009), while SGs are dynamic, liquid-like compartments with rapid turnover of residing RBPs (Kedersha *et al*, 2000, 2005; Mollet *et al*, 2008; Kroschwald *et al*, 2015; Patel *et al*, 2015). To determine how such stable and dynamic structures co-assemble into one compartment, we first determined the order of assembly by live imaging of heat-stressed cells. We observed that SGs formed rapidly after exposure to heat stress (after 10–30 min). Initially, these SGs were not visibly enriched for mutant SOD1 (Fig 2A and B). However, with prolonged stress (60–120 min), mutant SOD1 started to accumulate in SGs (Fig 2A and B). Simultaneously, the RBPs FUS and G3BP1 were partially depleted from SOD1-containing SGs (Fig EV2A and B). This shows that misfolded proteins accumulate in SGs and that the composition of SGs changes with time.

Next, we tested whether misfolded SOD1 forms a protein aggregate inside SGs. To probe the dynamic behavior of SOD1 in SGs, we used FRAP (fluorescence recovery after photobleaching). We observed that the mobile fraction of SOD1 in SGs was drastically reduced after prolonged stress (Fig 2C and D). This reduction of SOD1 mobility was only observed in SGs that showed high levels of SOD1 enrichment (Appendix Fig S3). This suggests that misfolded SOD1 becomes trapped in SGs as a result of aggregation. This is in direct contrast to the behavior of RBPs, which normally show dynamic behavior and rapid exchange between cytosol and SGs (Kedersha *et al*, 2000, 2005; Mollet *et al*, 2008; Patel *et al*, 2015). To directly compare the behavior of RBPs and misfolded SOD1 in the same SG, we photobleached both fluorophores simultaneously. We observed that FUS, as well as G3BP1, were relatively mobile. However, in the same SGs, SOD1(A4V) was immobile (Fig 2E–G). This suggests that misfolded SOD1 forms aggregates inside SGs, and that two SG components, FUS and G3BP1, retain their dynamic behavior, at least initially.

Considering that aberrant SGs contain two components with drastically different mobilities (RBPs and misfolded proteins), we wondered about the spatial organization of these SGs. In our imaging experiments, we noticed that the misfolded proteins sometimes

appeared to have a non-homogeneous distribution within SGs. However, this observation was difficult to interpret due to the resolution limit of conventional microscopy. Therefore, we performed structured illumination microscopy to obtain super-resolution images of aberrant SGs. Strikingly, we observed that misfolded SOD1 forms distinct domains, often at the periphery of SG and largely excluded from the areas containing RBPs (Fig 2H and I, Movie EV2, Appendix Fig S4), suggesting that the two materials are close to each other but spatially segregated within SGs. Interestingly, these SOD1 domains form only with prolonged stress when SOD1(A4V) is highly enriched in SGs (Fig EV2C and D). Together, these data suggest that misfolded proteins accumulate in SGs over time and form protein aggregates, which might affect the dynamics and functionality of SGs.

## Recruitment of misfolded proteins alters the dynamic properties of SGs

We have shown that after two hours of heat stress, only a fraction of SGs were highly enriched for misfolded SOD1 ("SOD1-positive SGs", defined as SGs that show strong enrichment of SOD1 over the cytosolic levels), while other SGs did not contain high levels of misfolded SOD1 ("SOD1-negative SGs", defined as SGs that do not show an enrichment of SOD1 over the cytosolic levels; Fig 1G, Appendix Fig S2B). Both types of SGs contained poly(A) mRNA (Fig EV3A) and did not vary dramatically in shape and size (Appendix Fig S5). In most cases, SGs in one cell had similar levels of SOD1 enrichment, suggesting changes on the global cellular level that affect the ability to regulate misfolded proteins in SGs. We decided to exploit this variability to test whether the accumulation of misfolded proteins affects the dynamic properties of SGs. Using live-cell microscopy, we observed that SOD1-negative SGs were dynamic, changing shape, and undergoing fusion and fission events (Fig EV3B–D). SOD1-positive SGs instead were less dynamic and showed a much lower frequency of fusion and fission events (Figs 3A and B, and EV3E). In few cases, we observed two distinct populations of SGs in a single cell (Fig 3C). In these cells, the SOD1-negative SGs were again more dynamic, changing shape, and fusing, while the SOD1-positive SGs were less dynamic and retained their shape for long periods of time (Fig 3C and D, Movie EV3). These results suggest that SOD1-positive SGs do not exhibit typical liquid-like properties of SGs, and this difference is often, but not always, caused by the specific physiological state of a cell.

To test whether these differences are also reflected at the molecular level, we performed FRAP experiments on G3BP1, a key component of SGs. Indeed, we observed a significantly reduced mobile fraction of G3BP1 in SOD1-positive SGs compared to SOD1-negative SGs (Fig 3E and F), suggesting that aggregation of misfolded proteins in SGs affects the mobility of key SG proteins such as G3BP. This might be caused by a shift from transient interactions to more stable interactions. It has been reported that free mRNA is required for SG formation and integrity, indicating the importance of RNA-based interactions in SGs (Kedersha *et al*, 2000; Bounedjah *et al*, 2014). In agreement with this, we found that microinjection of RNase A into heat-stressed cells expressing G3BP1-mCherry leads to a dramatic reduction of SG size within seconds (data not shown). We then used this

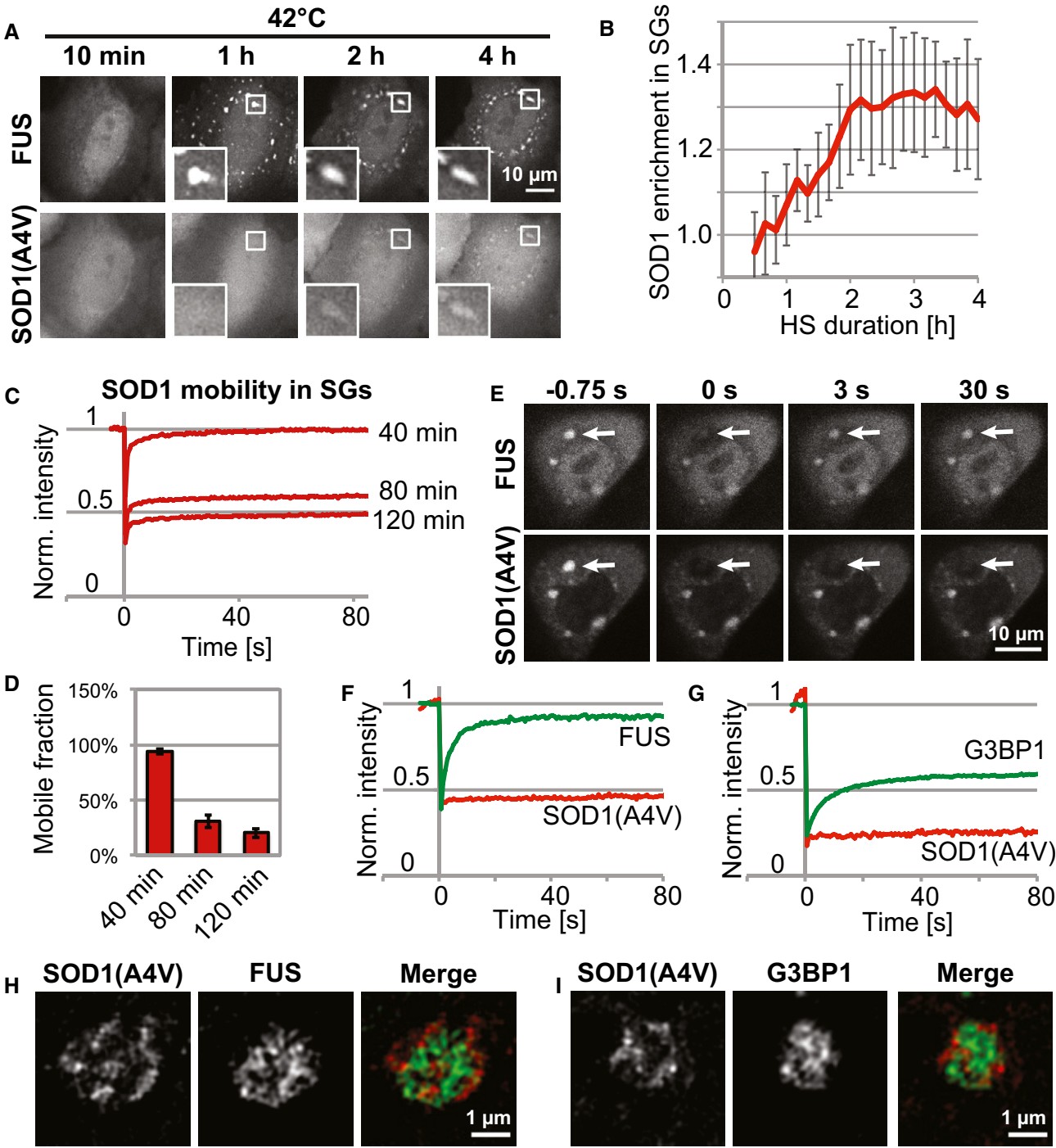

**Figure 2.  Misfolded proteins form aggregates after accumulating in SGs.**

A    Recruitment of FUS-mCherry and SOD1(A4V)-GFP into SGs in living cells exposed to heat stress.

B    Quantification of SOD1(A4V) enrichment in the SGs from (A) (intensity in SG/intensity outside SG) at different time points of heat stress. Mean values are shown (13–27 SGs in each frame). Error bars = standard deviation.

C    FRAP analysis of SOD1(A4V)-GFP in SGs after 40, 80, or 120 min of heat stress. Mean values are shown (8–13 SGs per condition).

D    Mobile fraction of SOD1(A4V) calculated from the FRAP analysis in (C). Error bars = SEM (8–13 SGs per condition).

E    SOD1-positive SGs were induced with heat stress (2 h) and photobleached (arrows) in two channels. Fluorescence recovery of FUS-mCherry, but not SOD1(A4V)-GFP, was observed.

F    Two-channel FRAP analysis of FUS-mCherry and SOD1(A4V)-GFP in SGs following a 2-h heat stress. Mean values are shown (5 SGs).

G    Two-channel FRAP analysis of G3BP1-mCherry and SOD1(A4V)-GFP in SGs following a 2-h heat stress. Mean values are shown (12 SGs).

H    Super-resolution image of a SG containing FUS-mCherry and SOD1(A4V)-GFP following a 2-h heat stress.

I    Super-resolution image of a SG containing G3BP1-mCherry and SOD1(A4V)-GFP following a 2-h heat stress.

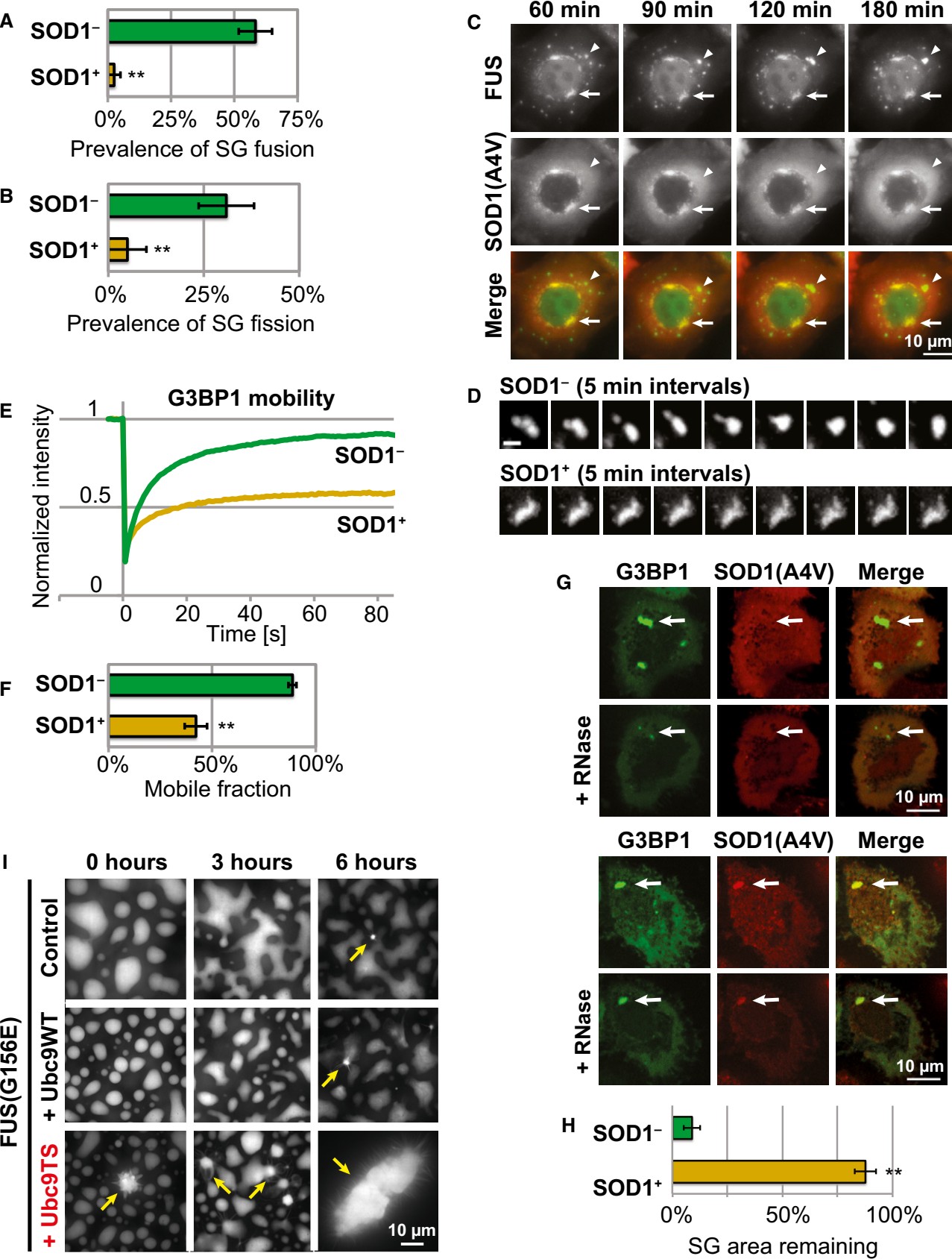

**Figure 3.**

**Figure 3.  SGs that accumulate misfolded proteins show aberrant behavior.**

A   Prevalence of SG fusion in cells with SOD1-negative SGs (SOD1$^-$) and cells with SOD1-positive SGs (SOD1$^+$) during 2-h recovery from heat stress (2 h). Cells express FUS-mCherry and SOD1(A4V)-GFP. Only cells with SGs persisting for 2 h were analyzed. Average from five experiments is plotted. Error bars = SEM. **$P < 0.01$ (*t*-test).

B   Prevalence of SG fission in cells with SOD1-negative SGs (SOD1$^-$) and cells with SOD1-positive SGs (SOD1$^+$) during 2-h recovery from heat stress. Average from five experiments is plotted. Error bars = SEM. **$P < 0.01$ (*t*-test).

C   A cell containing two populations of SGs, SOD1-negative (arrowheads) and SOD1-positive (arrows). Time indicates duration of recovery from heat stress (2 h). See also Movie EV3.

D   SGs from the representative cell shown in (C). SOD1-negative SGs undergo fusions and change shape. SOD1-positive SGs are less dynamic. Scale bar = 2 μm.

E   FRAP analysis of G3BP1-mCherry in cells co-expressing SOD1(A4V)-GFP. Following a 2-h heat stress treatment, G3BP1 was photobleached in SOD1-negative SGs or SOD1-positive SGs in the same sample. Mean values from ≥ 10 SGs for each category are plotted.

F   Mobile fraction of G3BP1 in SGs calculated from the FRAP analysis in (E). Error bars = SEM. **$P < 0.01$ (*t*-test).

G   Cells expressing G3BP1-mCherry and SOD1(A4V)-GFP were heat-stressed for 2 h. RNase A was then microinjected into cells with SOD1-negative SGs (upper panel) or cells with SOD1-positive SGs (lower panel) in the same sample. Images were taken immediately before injection and after injection (+RNase). Arrows indicate SGs.

H   Quantification of SG area remaining after RNase microinjection (four cells from each category). Error bars = SEM. **$P < 0.01$ (*t*-test).

I   Effect of misfolded proteins on *in vitro* reconstituted FUS compartments. FUS(G156E)-GFP was incubated either alone (control) or with purified Ubc9WT or Ubc9TS for the indicated time. In control samples, FUS phase-separated into droplets wetting the surface. In samples containing Ubc9TS, morphologically distinct particles with emanating fibers were prevalent (arrows).

approach in cells co-expressing SOD1(A4V) to compare the RNase sensitivity of SOD1-negative and SOD1-positive SGs. As expected, RNase injection led to rapid dissolution of SOD1-negative SGs, with only ~9% of the SG area remaining (Fig 3G, upper cell, and H). These remaining foci might correspond to the SG cores that were recently reported (Jain *et al*, 2016). Strikingly, RNase injection had a minimal effect on SOD1-positive SGs, with ~88% of the SG area remaining (Fig 3G, lower cell, and H). This suggests that the integrity of aberrant SGs does not depend on RNA-based interactions, but rather on more stable protein-protein interactions, which might directly involve misfolded proteins. Collectively, these data indicate that aberrant SGs are more stable, less dynamic, and less liquid-like than SGs devoid of misfolded proteins.

Next, we tested whether the presence of misfolded proteins can directly alter the properties of SGs in a simplified *in vitro* system. We reconstituted phase-separated liquid FUS compartments *in vitro* using an ALS-linked variant of FUS (G156E) that is more prone to undergo liquid-to-solid phase transition. We incubated FUS(G156E) either alone or in presence of Ubc9WT or Ubc9TS, and we monitored the morphological changes of FUS compartments over time. FUS compartments appeared spherical in solution (data not shown) and wetted the surface upon contact (Fig 3I). In the presence of Ubc9WT, FUS compartments were indistinguishable from FUS-only samples, suggesting that the wild-type protein does not have a major effect on FUS compartments *in vitro*. In contrast, we observed morphologically distinct particles forming upon mixing FUS with Ubc9TS (Fig 3I, yellow arrow, bottom left), suggesting that the

misfolded variant immediately alters the properties of FUS. Importantly, more inclusion-like particles and structures with fibers emanating from them formed during long-term incubation with Ubc9TS. These aberrant structures were also found in control samples, but at later time points and to a much lesser extent. This demonstrates that Ubc9TS can trigger a liquid-to-solid transition of FUS compartments *in vitro*. In conclusion, our data suggest that the presence of misfolded proteins promotes a liquid-to-solid phase transition of RNP granules.

## Chaperones are specifically recruited to aberrant SGs

To counter the misfolding and aggregation of proteins, cells have evolved chaperones and degradation systems such as the ubiquitin-proteasome machinery. We reasoned that chaperones might recognize the SGs that contain misfolded aggregated proteins. Indeed, we found that SOD1-positive SGs often contain ubiquitin and chaperones such as HSP27, HSP70, and VCP (Fig 4A). Again, we observed variability in the composition of SGs (note that we always define SGs by one of the SG markers such as FUS). We noticed that chaperones are mainly localized to SOD1-positive SGs and not to SOD1-negative SGs (Fig 4B, compare upper and lower panels). In cells with SOD1-negative SGs, the misfolded SOD1 and chaperones often co-localized in foci distinct from SGs (Fig 4B, lower panel, arrow). To confirm this observation, we used our automated image analysis assay to quantify the relative enrichment of proteins in individual SGs (Fig 4B). We find that SGs enriched for ubiquitin, HSP27 or HSP70, also tend to be highly

**Figure 4.  Chaperones are specifically recruited into SOD1-positive SGs.**

A   HeLa cells expressing SOD1(G93A)-GFP were heat-stressed for 2 h, fixed, and stained with antibodies against eIF3η, poly-ubiquitin, HSP27, HSP70, or VCP.

B   Examples from the imaging assay used to quantify protein enrichment in SGs. Automatic segmentation of SGs is based on FUS-mCherry signal. High enrichment of SOD1 and HSP27 in one SG is reflected by high fluorescence ratios (upper panel). Low intragranular levels of SOD1 and HSP27 are reflected by low fluorescence ratios (lower panel). In this cell, SOD1 and HSP27 localize to different foci (arrows) instead of accumulating in SGs. Note that SGs are always defined by the presence of FUS.

C   Variability in SG composition. HeLa cells expressing FUS-mCherry and SOD1(A4V)-GFP were heat-stressed for 2 h, fixed, and stained with antibodies against polyubiquitin, HSP27 or TDP-43. An automated imaging assay was used to quantify the enrichment of proteins in SGs (1,000 SGs plotted). The Pearson's correlation coefficient (*r*) is shown.

D   Temporal changes in SG composition. HeLa cells expressing FUS-mCherry were heat-stressed (HS) for 30, 60, 90, 120, or 150 min, fixed, and stained with antibodies against poly-ubiquitin, HSP27 or TDP-43. An automated imaging assay was used to quantify the percentage of SGs highly enriched for ubiquitin, HSP27 or TDP-43, at given time points (using a fluorescent ratio 1.5 as threshold). Mean values from three independent experiments are shown, each sample (one replicate of one time point) with > 400 SGs (average 2,836). Error bars = SEM.

                                    

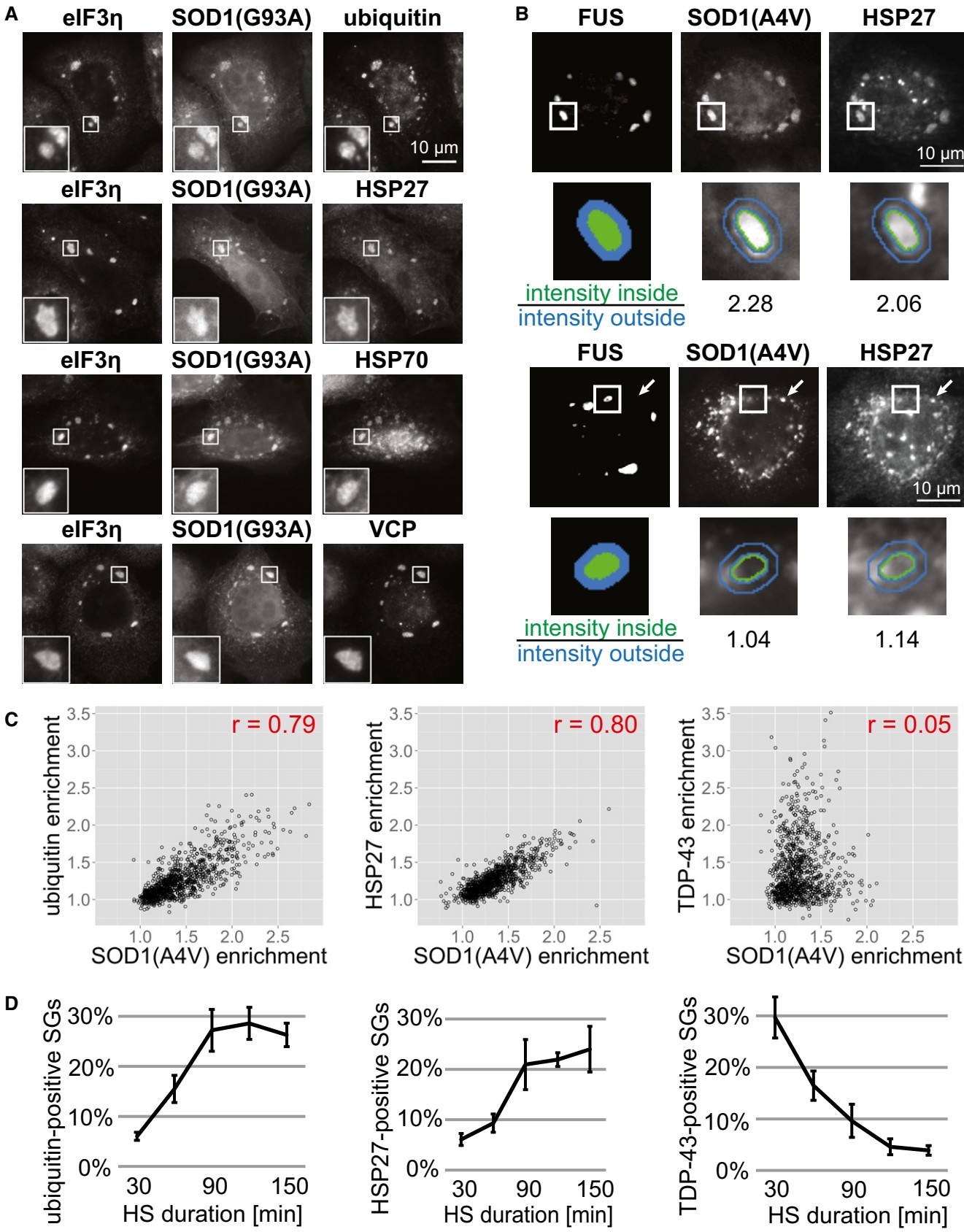

**Figure 4.**

enriched for misfolded SOD1 (Fig 4C, Appendix Fig S6). As a control, we used the RBP TDP-43. Although we observed some variability in TDP-43 enrichment in SGs, it did not show a strong correlation with the enrichment of SOD1 in SGs (Fig 4C). This clearly demonstrates that SOD1-positive SGs contain higher levels of chaperones, such as HSP27 and HSP70, suggesting that these chaperones are specifically recruited to aberrant SGs.

To understand the interplay between misfolded proteins and chaperones in SGs in more detail, we explored the temporal changes in SG composition. We used our image analysis assay to determine the composition of SGs at different time points during heat stress. As shown above in Fig 2A and B, misfolded SOD1 is initially not present in SGs, but accumulates in SGs with time. Using cells that did not express mutant SOD1, we observed that after short stress (30 min), SGs are devoid of poly-ubiquitin (Fig 4D). However, after longer stress (90 min and more), SGs enriched for poly-ubiquitin become more prevalent (Fig 4D). These data show that poly-ubiquitin accumulates in SGs over a similar time frame as misfolded SOD1 (Figs 4D and 2B), suggesting that there are endogenous misfolded proteins that are recruited to SGs with similar kinetics as mutant SOD1. Importantly, the small heat shock protein HSP27 also accumulated in SGs with time (Fig 4D). We noticed slightly delayed kinetics of HSP27 accumulation compared to poly-ubiquitin, suggesting that HSP27 is recruited to SGs in response to the accumulation of misfolded proteins. Interestingly, our control protein TDP-43 is depleted from SGs during prolonged heat stress (Fig 4D). This is in agreement with data presented in Fig EV2A and B, where the RBPs FUS and G3BP2 were partially depleted from SGs with prolonged stress. Thus, we conclude that the composition of SGs changes over time, with gradual accumulation of misfolded proteins, accompanied by the recruitment of chaperones and a depletion of some RBPs.

Based on these results, we reasoned that there must be endogenous misfolding-prone proteins that are normally targeted by the protein quality control machinery of SGs. Indeed, recent findings show that SGs have a tendency to sequester defective ribosomal products (DRiPs; Seguin *et al*, 2014; Ganassi *et al*, 2016). DRiPs are a heterogeneous group of proteins that are released from disassembling polysomes, which includes prematurely terminated polypeptides and nascent polypeptides. DRiPs are highly aggregation-prone, subject to protein quality control, and thus normally do not accumulate in SGs. However, when the protein quality control machinery is compromised, they accumulate and aggregate in SGs (Seguin *et al*, 2014; Ganassi *et al*, 2016). To determine whether DRiPs also accumulate under our experimental conditions in SGs, we labeled DRiPs through O-propargyl-puromycin (OP-puro) treatment of HeLa cells. This treatment leads to the termination of protein synthesis and generates puromycylated truncated proteins that are released from ribosomes. Indeed, we found that puromycylated DRiPs accumulate in heat-inducible SGs (Appendix Fig S7A). Interestingly, the OP-puro signal was present in SGs already after 30 min of heat stress (Appendix Fig S7A), when SGs are mostly devoid of ubiquitin or misfolded SOD1 (Figs 4D and 2B). This indicates that highly aggregation-prone DRiPs are already accumulating in newly forming SG at a very early stage. Surprisingly, in contrast to the gradual accumulation of ubiquitin and SOD1 in SGs, the fraction of OP-puro-positive SGs was decreasing with time (Appendix Fig

S7B). This indicates that SGs accumulate different classes of misfolded proteins with different kinetics. We do not know the reasons for this variability, but we speculate that DRiPs are most strongly recruited at the onset of stress, because translation rates are still high, whereas other misfolded proteins accumulate increasingly with persisting stress. Thus, we propose that stressed cells accumulate DRiPs and other misfolding-prone proteins and that aggregation of these proteins inside SGs leads to recruitment of chaperones to SGs.

## HSP70 prevents formation of aberrant SGs and promotes SG disassembly

We have shown that the chaperones HSP27 and HSP70 are specifically recruited to aberrant SGs that contain misfolded proteins. We thus speculated that chaperones regulate the interaction between SG components and misfolded proteins. To test whether these chaperones function in SG quality control, we used VER-155008 (VER), a chemical inhibitor of HSP70, combined with our high-content imaging assay to quantify changes in SG composition. We found that inhibition of HSP70 significantly increases the fraction of SGs that contain misfolded SOD1, Ubc9TS, or poly-ubiquitinated proteins (Fig 5A). This suggests that HSP70 acts to prevent the accumulation of misfolded proteins in SGs.

We observed that SGs induced by arsenate stress or proteasome inhibition are generally devoid of poly-ubiquitinated proteins (Fig 5B). We speculated that under these conditions, accumulation of misfolded proteins in SGs is prevented by the action of HSP70. Indeed, co-treatment of HeLa cells with VER and sodium arsenate or MG132 resulted in widespread appearance of ubiquitin-positive SGs (Fig 5B). Therefore, HSP70 is required to keep SGs free of misfolded proteins under diverse stress conditions.

Previous studies showed that HSP70 is required for the disassembly of SGs in budding yeast (Cherkasov *et al*, 2013; Walters *et al*, 2015). In agreement with these results, we observe that the disassembly of heat stress-induced SGs is much slower in HeLa cells treated with VER (Fig 5C and D). Given that HSP70 prevents the accumulation of misfolded proteins in SGs and promotes SG disassembly, we hypothesized that SG disassembly is hindered by the presence of misfolded proteins in SGs. To test this hypothesis, we used live-cell imaging to measure the rate of SG disassembly in cells containing SOD1-positive or SOD1-negative SGs. We observed that SG disassembly was significantly slower in cells that contained SOD1-positive SGs (Fig EV4A). This indicates that co-assembly with misfolded proteins delays the disassembly of SGs and leads to the persistence of SGs. We conclude that the chaperone HSP70 prevents the formation of aberrant SGs and enables rapid SG disassembly in the recovery phase. Thus, SG surveillance by chaperones is critical to maintain the normal composition and dynamism of SGs.

## Aberrant SGs can be targeted to the aggresome for degradation

Normally, SGs are transient compartments that disassemble when the stress subsides. However, impaired protein quality control through inhibition of HSP70 caused the persistence of SGs. Protein aggregates formed by misfolded proteins such as SOD1 are known to be stable and difficult to dissolve (Chattopadhyay & Valentine,

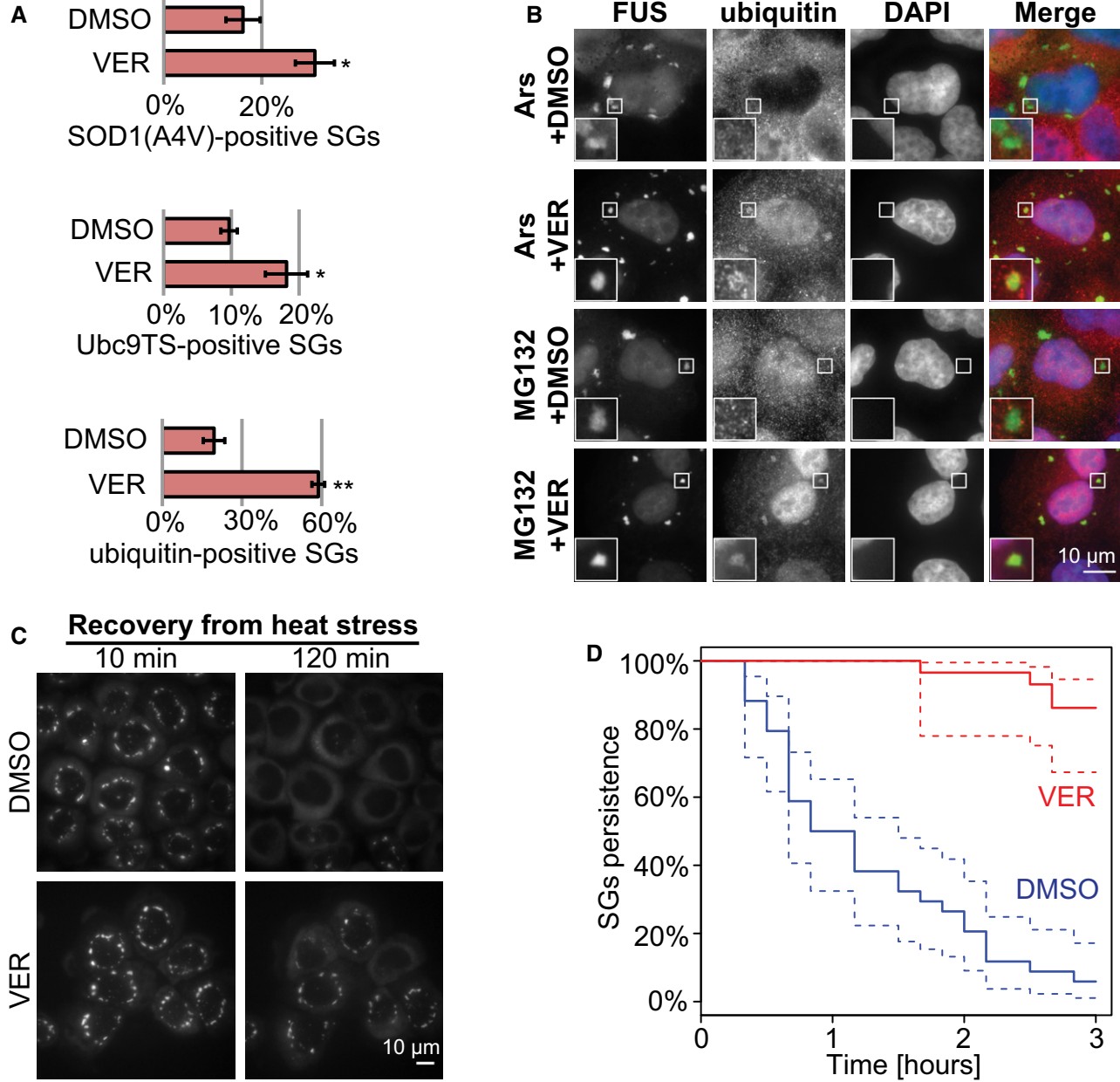

**Figure 5. HSP70 prevents formation of aberrant SGs and promotes SG disassembly.**

A   Fraction of SGs enriched for SOD1(A4V), Ubc9TS, or poly-ubiquitin. Cells were heat-stressed for 2 h in the presence of DMSO or 40 μM VER followed by fixation. SOD1(A4V)-GFP was co-expressed with FUS-mCherry, Ubc9TS-mCherry was co-expressed with G3BP2-GFP, and poly-ubiquitin was detected by immunofluorescence in G3BP2-GFP-expressing cells. Automated imaging assay was used with fluorescent ratio threshold 1.4. Mean values from three experiments are shown, each with > 300 SGs (average 2,028). Error bars = SEM. *P < 0.05, **P < 0.01 (t-test, compared to DMSO).

B   HeLa cells expressing FUS-mCherry were treated for 3 h with 10 μM MG132 or 2 h with 1 mM sodium arsenate and co-treated with 40 μM VER or DMSO. Cells were fixed and stained for poly-ubiquitin. Ubiquitin is visible in SGs in the conditions containing VER. Ubiquitin is also enriched in the nucleus in conditions where cells were treated with VER and/or MG132, both of which impair proteostasis, presumably causing accumulation of ubiquitinated proteins in the nucleus.

C   HeLa cells expressing G3BP2-GFP were heat-stressed for 2 h in the presence of DMSO or 40 μM VER and then imaged at 37°C for indicated time.

D   Quantification of SG persistence from live-cell imaging as described in (C). Complete SG disassembly was scored in cells treated with DMSO (34 cells) or VER (29 cells) and plotted using survival analysis in R. Dashed lines = 95% confidence intervals.

2009). To determine whether aberrant SGs can be disassembled at all, we followed them by fluorescence live-cell imaging during recovery from heat stress. Interestingly, we observed that SGs containing misfolded SOD1 could be completely disassembled within 2–3 h, with both RBPs and misfolded proteins leaving the SGs with time (Fig EV4B). In general, we observed that the RBPs disassembled faster than the misfolded proteins previously contained in SGs, suggesting a disassembly mechanism that first removes the more mobile RBP components from aberrant SGs and then disassembles more solid misfolded proteins.

However, we noticed variability from cell to cell, with some cells in which the SGs persisted for longer times. In these cells, SGs were slowly transported toward a large perinuclear inclusion, which contained high amounts of the misfolded proteins Ubc9TS or SOD1(A4V) (Figs 6A and EV5A, and Movie EV4). This appeared to be a directional transport toward one point, presumably the microtubule-organizing center (Fig 6B). The structures were reminiscent of the aggresome, a perinuclear inclusion that sequesters misfolded proteins by directional transport along microtubules (Johnston *et al*, 1998; García-Mata *et al*, 1999). Indeed, formation of the inclusions could be disrupted by treatment with the microtubule-disassembling drug nocodazole, suggesting that it is dependent on microtubule transport (Fig 6C). Aggresomes are believed to target misfolded proteins for degradation by autophagy (Fortun *et al*, 2003; Taylor *et al*, 2003; Iwata *et al*, 2005; Gamerdinger *et al*, 2011). To characterize these inclusions, we performed immunofluorescence staining of cells recovering from heat stress. After 6 h of recovery, most cells were devoid of SGs. However, in some cells, small SGs could be found at the aggresome containing the autophagy receptor p62 (Fig 6D, insets). These inclusions were also surrounded by a vimentin cage, a typical feature of aggresomes (Fig 6D, arrows; Johnston *et al*, 1998). These results demonstrate that persistent SGs can be transported along microtubules to the aggresome.

To test whether these persistent SGs are eventually degraded by autophagy, we used the autophagy inhibitor wortmannin and quantified the frequency of SG localization at the aggresome after 6 h of recovery. At this time point, treatment with wortmannin did not significantly affect the percentage of cells containing SGs or cells containing an aggresome (Fig 6E). However, the treatment with wortmannin significantly increased the percentage of cells that contained SGs associated with the aggresome (Fig 6E). This indicates that autophagy is used to specifically degrade the persistent SGs that undergo transport to the aggresome. However, autophagy does not seem to be the preferred pathway of SG clearance, as most SGs disassemble rapidly after stress and persisting SGs continuously decrease in size when they are transported to the aggresome (Movie EV4 and Fig EV5A), suggesting that they are subject to chaperone-mediated disassembly.

Aggresomes have been studied mostly under conditions of proteasome inhibition (Johnston *et al*, 1998, 2000; Fortun *et al*, 2003; Kawaguchi *et al*, 2003; Corcoran *et al*, 2004; Kaganovich *et al*, 2008; Yung *et al*, 2015). However, we were able to consistently induce aggresome formation by 2-h heat stress followed by a 6-h recovery period. Interestingly, we found that several proteins involved in protein quality control, such as ubiquitin, HSP27, HSP70, and VCP, were localized to SGs after heat stress, but redistributed to the aggresome during recovery from heat stress (Fig EV5B). In both situations, the proteins colocalize with misfolded Ubc9TS. This suggests that there is a transport of material from SGs to the aggresome, either as aggregated proteins or in the form of aberrant SGs. Collectively, these results demonstrate a direct connection between SGs and the aggresome, which has not been reported so far.

Our data suggest that aberrant SGs can be cleared through two pathways. A fast pathway (predominant in our conditions) involves disassembly of aberrant SGs by chaperones such as HSP70. A slow pathway involves transport of persistent SGs along with misfolded proteins to the aggresome, a large protein inclusion that is eventually degraded by autophagy.

## Discussion

In this paper, we report a complex interplay between SGs and misfolded proteins in stressed human cells. We show that misfolding-prone proteins, such as ALS-associated variants of SOD1, accumulate in SGs and change their material properties, composition, and dynamic behavior, thus revealing a previously unrecognized heterogeneity of RNP granules in stressed human cells. We find that cells employ a system of chaperone-mediated surveillance to monitor the composition of SGs and prevent a conversion into an aberrant state. Chaperone action is further required to disassemble aberrant SGs. Finally, we characterize a system of SG transport to the aggresome, which serves to selectively degrade persisting SGs (Fig 7). This suggests that chronic stress and failure to manage misfolded proteins triggers the conversion of SGs into an aberrant state, which may have relevance for diseases such as ALS.

SGs are highly dynamic structures that sequester RNAs. Only very recently, it has become clear how they form. The current thinking is that this involves a process known as liquid–liquid phase separation (Brangwynne *et al*, 2009, 2015; Li *et al*, 2012; Hyman *et al*, 2014; Molliex *et al*, 2015; Patel *et al*, 2015). In liquid–liquid phase separation, an initially homogeneous solution of RBPs and RNAs demixes into two distinct liquid phases that then stably coexist. One of these phases is enriched for RNAs and RBPs and forms a compartment, which allows diffusion of molecules inside, but is separated from the surrounding milieu by a boundary. Importantly, phase-separated compartments are metastable and convert with time from a liquid to a solid aggregated state, which is reminiscent of pathological aggregates found in patient cells (Lin *et al*, 2015; Molliex *et al*, 2015; Murakami *et al*, 2015; Patel *et al*, 2015). This raised several important questions, which so far have remained unanswered: How is the physiological state of RNP granules maintained during long periods of stress? How is a pathological conversion of RNP granules triggered in cells? Are protein aggregates nucleated inside phase-separated RNP compartments, and if so, is there a subsequent maturation process that leads to the formation of pathological aggregates?

Our study now provides important insights into these questions. We show that SGs form rapidly after exposure to stress and are initially highly dynamic and liquid-like, but then begin to sequester misfolded proteins (Fig 2A, Movie EV5). We find that misfolded proteins form aggregates in SGs (Fig 2) and that this affects the properties of SGs, including their RNase sensitivity and the mobility of key SG proteins such as G3BP1 (Fig 3). Interestingly, a recent study demonstrated that ALS-linked mutant SOD1 protein directly interacts with G3BP1 (Gal *et al*, 2016). Thus, we speculate that misfolded proteins such as SOD1 interact with RBPs inside SGs, and that these aberrant interactions trigger a conversion of RBPs into a non-dynamic, aggregated state. Interestingly, we observe that addition of misfolded proteins to *in vitro* reconstituted FUS droplets causes immediate morphological changes, which is in agreement with the formation of a mixed assembly consisting of FUS molecules aberrantly interacting with aggregates of misfolded proteins (Fig 3I, yellow arrow in the bottom left panel).

Using a quantitative imaging-based approach, we observe a clear correlation between the amount of misfolded proteins and the amount of chaperones inside SGs (Fig 4C). Moreover, the amount of misfolded proteins and chaperones inside SGs increases steadily

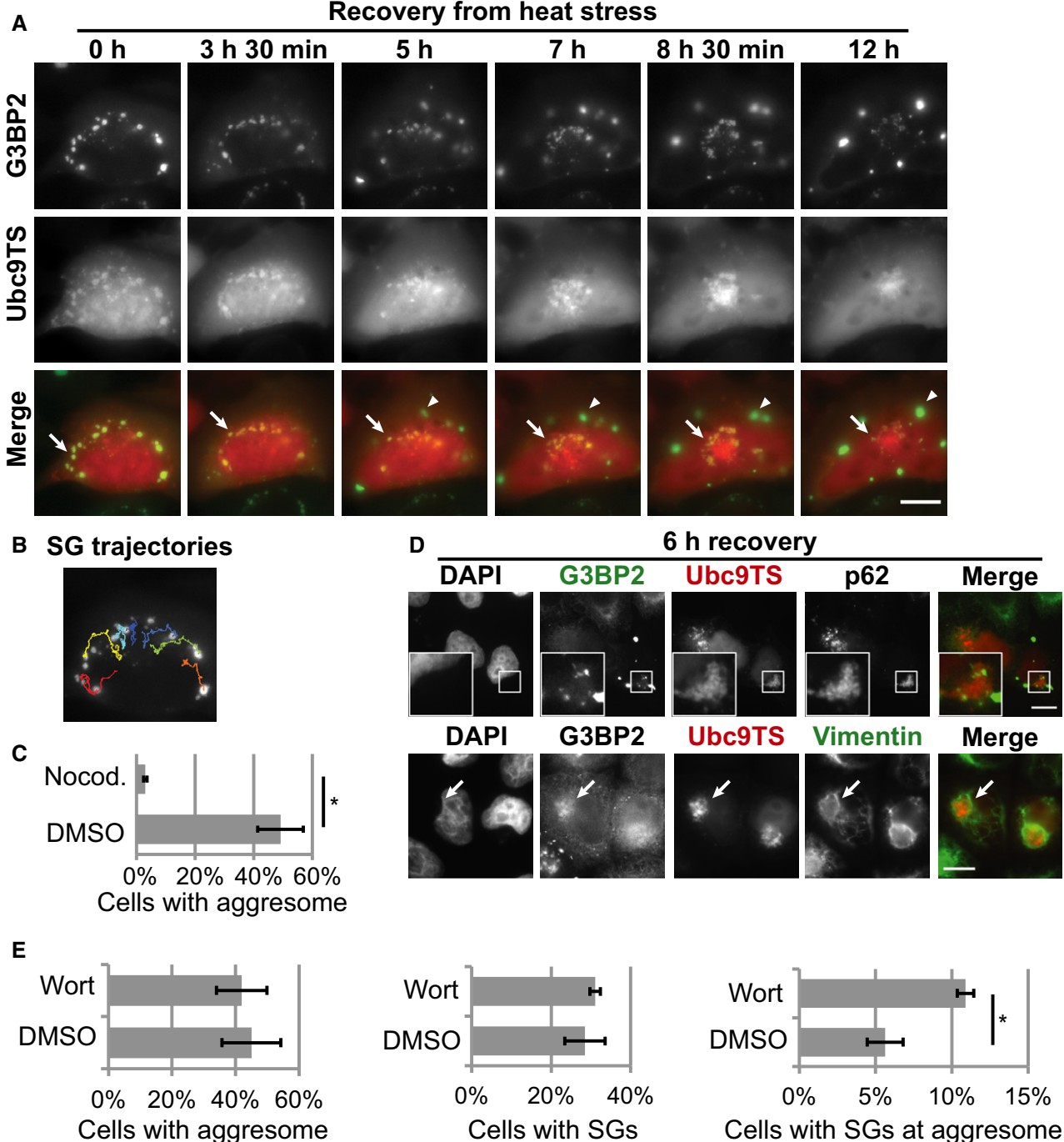

**Figure 6. Persisting aberrant SGs are transported to the aggresome for degradation.**

A   HeLa cells expressing G3BP2-GFP and Ubc9TS-mCherry were heat-stressed for 2 h and then imaged at 37°C. Time indicates duration of recovery. Ubc9TS-containing SGs (arrows) are transported from cell periphery toward the aggresome, where they slowly disappear. Meanwhile, Ubc9TS accumulates in the aggresome and new SGs devoid of Ubc9TS are formed (arrowheads). Scale bar = 10 μm. See also Movie EV4.

B   Trajectories of SGs from the cell depicted in (A).

C   Fraction of cells with an aggresome. HeLa cells expressing G3BP2-GFP and Ubc9TS-mCherry were treated with 5 μM nocodazole or DMSO, followed by 2 h of heat stress, 6 h of recovery at 37°C and fixation. Cells with a single large Ubc9TS inclusion were counted from at least 100 cells expressing Ubc9TS. Mean values from three experiments are shown. Error bars = SEM. *$P < 0.05$ (*t*-test).

D   HeLa cells expressing G3BP2-GFP and Ubc9TS-mCherry were treated with 2-h heat stress followed by 6 h recovery at 37°C. The cells were then fixed and stained for p62 or vimentin. Scale bars = 10 μm. Arrow indicates the position of aggresome surrounded by vimentin cage.

E   Cells expressing G3BP2-GFP and Ubc9TS-mCherry were treated with 1 μM wortmannin (Wort) or DMSO during heat stress (2 h) and subsequent recovery (6 h). After fixation, at least 100 cells expressing Ubc9TS were examined to determine the presence of aggresome, presence of SGs, and the localization of SGs at the aggresome region. Mean values from three experiments are shown. Error bars = SEM. *$P < 0.05$ (*t*-test).

    

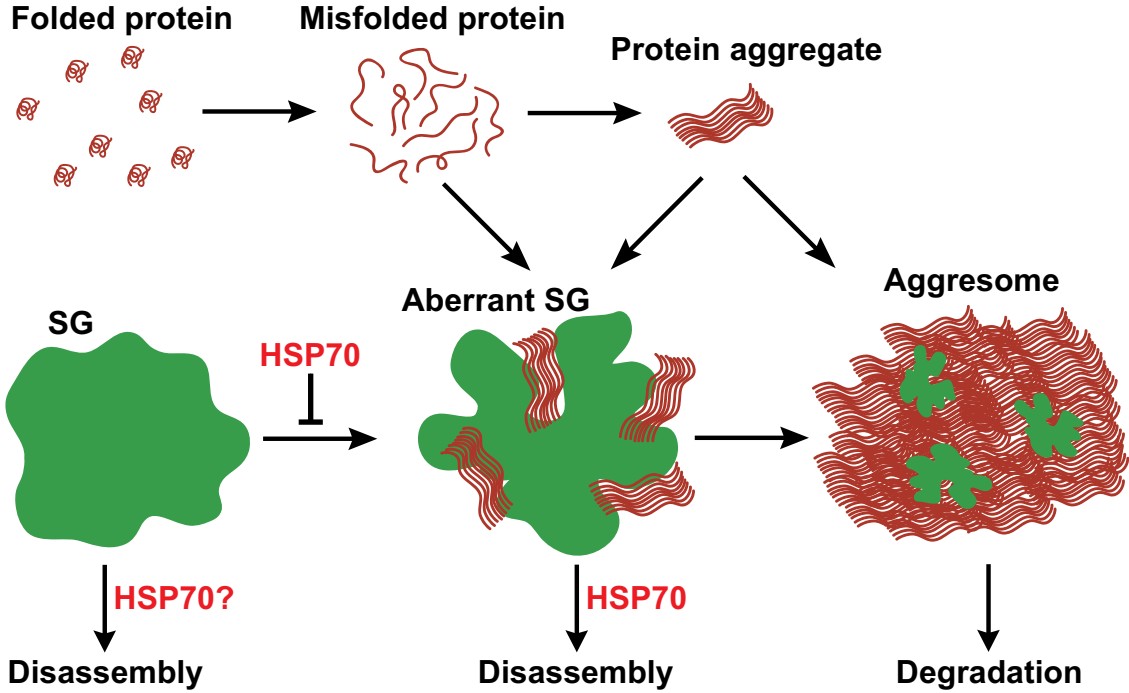

**Figure 7. Model illustrating the interplay between misfolded proteins and SGs.**
Misfolded proteins are generated due to mutations or stress conditions. These misfolded proteins can form aggregates that can assemble into large inclusions such as the aggresome. However, misfolded proteins can also form aggregates in SGs. The chaperone HSP70 prevents the accumulation of misfolded proteins in SGs. SGs that co-assemble with misfolded proteins are more stable, less dynamic, do not exhibit the typical liquid-like properties, and recruit chaperones. SG disassembly is promoted by HSP70, while persisting aberrant SGs are transported to the aggresome and targeted for degradation by autophagy.

with continuous stress (Figs 2B and 4D). This is consistent with a slow maturation process, where misfolded proteins become trapped inside SGs and then recruit chaperones and other PQC factors. This suggests that heat-induced SGs remodel their composition and increasingly turn into structures with properties of protein aggregates. This also involves the release of some SGs components such as TDP-43 from aberrant SGs (Figs 4D and EV2A and B). We speculate that the capacity of the PQC system under continuous heat stress conditions may not be sufficient to remove misfolded proteins and promote complete disassembly of SGs. Thus, we conclude that the composition of SGs is highly variable in space and time, and that the heterogeneity of SGs is dependent on the duration of stress, and possibly other factors such as the capacity of the cellular PQC system.

Interactions of misfolded proteins with RBPs in so-called heat shock granules have previously been observed in yeast (Cherkasov *et al*, 2013; Kroschwald *et al*, 2015). Indeed, although these assemblies seem to be related to SGs, they have more solid properties than those of cultured mammalian cells (Kroschwald *et al*, 2015). In addition, yeast heat shock granules depend on a specific molecular chaperone for disassembly: the ATP-driven machine Hsp104. In previous studies, we and others used the model misfolding-prone protein luciferase to investigate the interplay between misfolded proteins and SGs (Cherkasov *et al*, 2013; Kroschwald *et al*, 2015). This protein shows extensive co-aggregation with heat stress granules in yeast, but forms aggregates that are separate from SGs in mammalian cells (Cherkasov *et al*, 2013; Kroschwald *et al*, 2015). We speculate that this may be due to different degrees of luciferase

unfolding in yeast and mammalian cells. Thus, we conclude that misfolded proteins differ in their ability to interact with SG components, and that Ubc9TS and mutant SOD1 may be thermodynamically less stable and thus undergo more extensive interactions with SGs. Future studies will have to determine the exact structural features that allow misfolded proteins to interact with RBPs in liquid membrane-less compartments.

Previous studies reported that mammalian SGs contain ubiquitin, autophagy machinery, and chaperones such as HSP27 and VCP (Kedersha *et al*, 1999; Kwon *et al*, 2007; Buchan *et al*, 2013; Seguin *et al*, 2014). However, the proteins that are targeted by these factors remained unknown. Our results and those of a recently published study (Ganassi *et al*, 2016) argue that defective ribosomal products and other misfolded proteins are the targets of ubiquitination, chaperones, and the autophagy machinery. Moreover, we show that these factors are not present in newly forming SGs, but rather appear progressively in a slow maturation process that goes along with a conversion of physiological SGs into aggregates. Importantly, we demonstrate that the chaperone HSP70 keeps SGs devoid of misfolded proteins and promotes SG disassembly in the recovery phase. Combined, these results show that chaperone-mediated disassembly is the preferred pathway of SG clearance in mammalian cells and that yeast (Kroschwald *et al*, 2015; Wallace *et al*, 2015) and mammalian cells favor recycling of RBPs rather than their degradation. However, we also find evidence for an additional pathway of SG clearance, which involves microtubule-based transport of SGs to the aggresome followed by autophagy-dependent degradation (Fig 6). Degradation of SGs by autophagy seems to preferentially

affect persisting SGs. We noticed that persisting SGs en route to the aggresome slowly decrease in size, suggesting that they are still dynamic and can at least partially dissolve (Movie EV4). We suspect that these SGs have a less dynamic core, which may result from extensive aberrant interactions with misfolded proteins. Indeed, a recent study proposed that SGs consist of a core and a shell structure (Jain *et al*, 2016). Because of their inability to dissolve, these solid core structures may be preferentially targeted by autophagy-mediated degradation.

Our findings may have implications for ALS pathogenesis. Our data show that mutant ALS-linked variants of SOD1 have a tendency to accumulate in SGs, a subcellular compartment that harbors other ALS-linked proteins such as FUS and TDP-43 (Robberecht & Philips, 2013). In our *in vitro* "aging" reaction (Fig 3I), we observe that misfolded proteins accelerate the conversion of FUS into an aggregated state in a similar manner as ALS-associated mutations in FUS (Patel *et al*, 2015). Previous studies have reported interaction between mutant SOD1 and SG components in cultured cells, mouse models, and patient samples (Lu *et al*, 2009; Sumi *et al*, 2009; Gal *et al*, 2016). However, the implications of these observations have remained mostly unclear. We speculate that misfolded proteins inside SGs may provide a template for the conversion of RBPs into an aberrant state. In agreement with this, we observe that misfolded proteins promote a rapid liquid-to-solid transition of reconstituted FUS granules *in vitro* (Fig 3I). To assess the significance of our findings for ALS, it will be necessary to explore the interplay between misfolded proteins and RNP granules in the context of motor neurons derived from ALS patients.

Aberrant SGs may also play important roles in other age-related diseases than ALS. SG components have been found to associate with pathological inclusions in several neurological disorders. Prion protein aggregates, for example, sequester mRNA (Goggin *et al*, 2008) and inclusions of mutant huntingtin colocalize with the SG component TIA-1 (Waelter *et al*, 2001). TIA-1 has also been shown to interact with Tau inclusions in mouse models of Alzheimer's disease and in post-mortem patient tissues (Vanderweyde *et al*, 2012). Interestingly, overexpression of TIA-1 stimulates the formation of Tau inclusions in SH-SY5Y cells, which led the authors to suggest that SG formation might stimulate tau pathophysiology (Vanderweyde *et al*, 2012). This shows that there is extensive cross-talk between misfolded proteins and RNP components. Therefore, it seems possible that SGs contribute to the formation of pathological aggregates in many neurodegenerative disorders, possibly by serving as a platform that initiates the aggregation of many additional proteins, thus setting in motion a vicious cycle of aberrant phase transitions and cellular decline.

# Materials and Methods

### Cell culture and cell lines

HeLa cells were cultured in 4,500 mg/l glucose DMEM supplemented with 10% fetal bovine serum and penicillin + streptomycin antibiotics (all Gibco Life Technologies). Cells were maintained at 37°C in a 5% $CO_2$ incubator. BAC recombineering technology (Poser *et al*, 2008) and fluorescence-activated cell sorting were used to generate stable HeLa cell lines expressing endogenous levels of

G3BP2-GFP, FUS-GFP, G3BP1-mCherry, or FUS-mCherry. GFP cell lines were cultured in the presence of Geneticin (Gibco, 250 µg/ml), and mCherry cell lines were cultured in the presence of blasticidin (Life Technologies, 5 µg/ml). Heat stress experiments were performed in a 5% $CO_2$ incubator at 43°C.

### Live-cell microscopy

HeLa cells were imaged using a DeltaVision imaging system equipped with SoftWorx software (Applied Precision) and based on Olympus IX71 microscope. The oil immersion objective used was 60×/1.42NA/UPlanSA. A stable temperature (37°C) was maintained during imaging. Cells were imaged in 3.5-cm glass bottom Petri dishes (MatTek) in $CO_2$-independent medium Leibovitz's L-15 (Gibco) or DMEM supplemented with 20 mM HEPES. For imaging during heat stress, a Warner heating chamber (Warner instruments) was used. Deconvolution was performed using SoftWorx (conservative algorithm, medium noise filtering). Maximum intensity projections from collected Z-stacks were generated in Fiji (Schindelin *et al*, 2012). Fiji was also used for brightness adjustment, cropping, creating scale bars and insets (3× zoom). MS Excel was used to plot the prevalence of SG fusion and fission. Trajectories of SGs were generated with the TrackMate plugin in Fiji.

### High-content imaging assay

Accumulation of proteins in SGs was quantified in cells fixed with 10% PFA for 10 min and mounted in DAPI-Fluoromount G (SouthernBiotech), using the Scan^R imaging platform (Olympus) based on Olympus IX81 microscope and equipped with 60×/1.35NA/UPlanSA oil immersion objective. Automated image acquisition was used to obtain at least 200 images in each sample. The data were imported into Scan^R Analysis software, where SGs were automatically segmented with the edge detection algorithm. The relative enrichment of proteins in each detected SG was calculated as a ratio of mean fluorescence intensity inside the SG divided by mean intensity in a region surrounding the SG. Thus, we quantify the local enrichment of a protein in SGs and the results are not directly affected by changes in the total amount of a protein in the cell. To define the percentage of SGs highly enriched for a particular protein, we counted the number of SGs with the relative enrichment above a threshold level. Mean percentage and SEM were calculated from three biological experiments and plotted using MS Excel. For analyzing correlations between enrichment levels of two proteins, we used R to generate scatter plots and calculate Pearson's correlation coefficients. Scan^R analysis was also used to measure SG area and circularity, defined as $4\pi(\text{area/perimeter}^2)$. For measuring protein enrichment in SGs in live cells, the images were acquired on Delta-Vision microscope (see above) and imported into Scan^R Analysis.

### Plasmids and transfections

SOD1-GFP plasmids were a gift from Elizabeth Fisher (Addgene # 26407, 26408, 26409, 26410, 26411; Stevens *et al*, 2010). Plasmids encoding Ubc9TS or Ubc9WT were a gift from Judith Frydman (Addgene # 20369, 20368; Kaganovich *et al*, 2008). The cDNA of Ubc9TS or Ubc9WT was cloned into a pcDNA3.1-mCherry "destination vector" using Gateway cloning. The plasmid encoding

mCherry-VHL (von Hippel-Lindau tumor suppressor) was a gift from Daniel Kaganovich. Plasmids were transfected into HeLa cells using Lipofectamine 2000 transfection reagent (Invitrogen) and Opti-MEM (Gibco).

**Drug treatments**

Where indicated, the following drugs were used: VER-155008 (Santa Cruz Biotechnology), MG132 (Sigma-Aldrich), nocodazole (Appli-Chem), wortmannin (Santa Cruz Biotechnology), sodium arsenate dibasic heptahydrate (Alfa Aesar).

**Immunofluorescence**

HeLa cells grown on coverslips were fixed using 4% PFA in PBS. Permeabilization was performed using 0.2% Triton X-100 in PBS for 10 min. 3% BSA in PBS was used for blocking. All antibodies were diluted in 1% BSA in PBS. The following antibodies were used: goat anti-eIF3η (N-20, Santa Cruz Biotechnology), mouse anti-polyubiquitin (FK1, Enzo Life Sciences), mouse anti-HSP27 (ADI-SPA-800, Enzo Life Sciences), mouse anti-HSP70 (ADI-SPA-810, Enzo Life Sciences), mouse anti-VCP (MA3-004, Thermo Fisher Scientific), mouse anti-p62/SQSTM1 (D-3, Santa Cruz Biotechnology), mouse anti-TARDBP (41-7.1, Santa Cruz Biotechnology), and mouse anti-vimentin (V9, Santa Cruz Biotechnology). Secondary antibodies conjugated with Alexa Fluor fluorophores (Invitrogen) were then applied, followed by washing in PBS. Coverslips were mounted on microscope slides in DAPI-Fluoromount G (SouthernBiotech). The cells were then imaged using the DeltaVision imaging system (Applied Precision) or the Scan^R imaging platform (Olympus), as described above.

**Fluorescence recovery after photobleaching (FRAP)**

FRAP measurements were performed on a spinning-disk confocal microscope (Olympus IX81). The oil immersion objective used was 60×/1.35NA/UPlanSA. Cells were imaged in 3.5-cm glass bottom Petri dishes (MatTek) in $CO_2$-independent medium Leibovitz's L-15 (Gibco) and at stable temperature (37°C). To photobleach SGs, a 20 ms laser pulse was applied to a 20 × 20 pixel (2.7 × 2.7 μm) square. Confocal images were obtained before and after photobleaching. Fiji was used to measure fluorescent intensities, and easyFRAP (Rapsomaniki *et al*, 2012) was used to generate normalized curves and calculate mobile fractions. Double normalization was used to correct for fluorescence loss during imaging and for differences in the starting intensity. Mobile fractions were determined from exponential equations fitted to individual FRAP curves. Normalized curves and mobile fractions were plotted using MS Excel.

**Super-resolution microscopy**

HeLa cells expressing FUS-mCherry or G3BP1-mCherry were grown on 170 μm selected coverslips (Menzel-Glaser) and transfected with SOD1(A4V)-GFP. Following a 2-h heat stress treatment, the cells were fixed using 4% PFA in PBS and mounted in Vectashield H-1000 (Vector Laboratories). Super-resolution microscopy was performed on DeltaVision OMX (Applied Precision) imaging system equipped with structured illumination and a 60×/1.42NA/UPlanSA

oil immersion objective. Optical sections with 125 nm spacing were acquired. Image reconstruction was performed by the SI Reconstruction function in SoftWorx (Applied Precision). Fiji was then used for cropping, contrast adjustment, and drawing of scale bars. For colocalization analysis, individual SGs were cropped in Fiji and analyzed in JACoP (Bolte & Cordelières, 2006). Pearson's correlation coefficient was calculated as a measure of colocalization in the SG. Costes' randomization (Costes *et al*, 2004) was used to calculate Pearson's coefficients for randomized images (1,000 randomization rounds, block size 120 × 120 nm). Pearson's coefficients for the original images were then compared to the coefficients for randomized images and plotted using R. 3D Viewer plugin in Fiji (Schmid *et al*, 2010) was used to generate 3D visualization of image stacks.

**O-Propargyl-puromycin labeling of nascent peptides**

Cells were incubated with 25 μM O-propargyl-puromycin (Jena Bioscience) for indicated time at 43°C and fixed using 4% PFA in PBS. As described in Liu *et al* (Liu *et al*, 2012), the cells were then washed with TBS, permeabilized with TBST (TBS with 0.2% Triton X-100), and washed with TBS again. CuAAC detection of the incorporated O-propargyl-puromycin was performed as previously described (Liu *et al*, 2012). Afterward, cells were washed in TBS and processed for immunofluorescence staining as described above.

**Protein purification**

FUS(G156E)-GFP was purified as described previously (Patel *et al*, 2015). Recombinant Ubc9WT-6xHis and Ubc9TS-6xHis were purified from One Shot® BL21 Chemically Competent *E. coli* (Thermo Fisher Scientific). Cells were lysed using EmulsiFlex-C5 (Avestin) in a lysis buffer containing 350 mM KCl, 50 mM HEPES/HCl pH 7.4, 20 mM imidazole, 2 mM DTT, and 1× EDTA-free protease inhibitor cocktail (Roche Applied Science). Ni-NTA resin (Qiagen) was used to trap Ubc9 from the supernatant of the lysate. After washing the column with lysis buffer, Ubc9 was eluted with elution buffer (lysis buffer + 250 mM imidazole). The eluate was immediately loaded onto the size-exclusion chromatography column HiLoad 16/600 Superdex 75 pg (GE Life Sciences) using either BioCad Perfusion Chromatography Workstation (Applied Biosystems) or Akta Ettan FPLC system (GE Life Sciences). The low-salt buffer used for the size-exclusion chromatography contained 50 mM KCl, 50 mM HEPES/HCl pH 7.4, 2 mM EDTA, and 2 mM DTT. Subsequently, the fractions containing Ubc9 were loaded onto cation exchange chromatography column HiTrap SP HP (GE Life Sciences) using either BioCad Perfusion Chromatography Workstation or Akta Ettan FPLC system. The protein was eluted by applying a tenfold linear gradient of salt—from 50 mM KCl to 500 mM KCl over 10 column volumes. The protein was then dialyzed in a low-salt buffer, flash-frozen, and the aliquots were stored at −80°C.

**Fluorescence spectroscopy**

Ubc9WT or Ubc9TS was diluted from stock solutions to a final concentration of 1 μM into 50 mM HEPES, 50 mM KCl, 2 mM EDTA, pH 7.4. Samples were incubated for 1 h and then transferred into fluorescence cuvette. Fluorescence spectra were obtained with FluoroMax 3 (HORIBA Jobin Yvon) equipped with a four-position

cuvette holder and water-bath (Haake) at 22°C. Fluorescence excitation was at 275 nm with a slit width of 5 nm. Fluorescence emission was recorded from 300 to 600 nm with 5-nm slits. For measuring surface hydrophobicity, 1 μM of Ubc9WT or Ubc9TS was incubated for 1 h with 100 μM ANS (1-anilinonaphthalene-8-sulfonic acid) in 50 mM HEPES, 50 mM KCl, 2 mM EDTA, pH 7.4. ANS fluorescence spectra were obtained with FluoroMax 3 (HORIBA Jobin Yvon). Excitation was at 370 nm, and fluorescence emission spectra were recorded from 390 to 600 nm.

### Fluorescence imaging of purified Ubc9 and FUS

For imaging of purified proteins, Ubc9 was labeled with Cy3 dye (Lumiprobe) using maleimide chemistry (cysteine coupling) and used in a mix with unlabeled Ubc9 (labeled:unlabeled = 1:10). This mix was titrated to 5 μM FUS-GFP in a buffer containing 75 mM KCl, 0.4% glycerol, 33 mM HEPES, 4.2 mM Tris, and 0.75 mM DTT. Images were taken on a spinning-disk confocal microscope (Olympus IX81). Fluorescence intensities of Ubc9 in the FUS droplets and in solution were measured at various concentrations of Ubc9 using Fiji. The fluorescence intensity ratios (Ubc9 signal inside the droplets/Ubc9 signal outside the droplets) were plotted.

### *In vitro* aging of FUS compartments

5 μM of purified Ubc9TS or Ubc9WT was incubated with equimolar amount of FUS(G156E)-GFP in 10 mM Tris/HCl, 70 mM KCl, 50 mM EDTA, and 1 mM DTT, pH 7.4. The reactions were incubated for indicated time in separate wells of a low protein binding 384-well plate (Greiner) and centrifuged at 800 rpm at RT. Plates were imaged using an Andor spinning-disk confocal microscope equipped with 100× oil immersion objective. 10-μm-thick *Z*-stacks were acquired, and maximum intensity projections were generated in Fiji. Representative images are shown.

### SDS–PAGE and Western blotting

For Western blot analysis, cells were harvested with a plastic cell scraper, mixed with Laemmli buffer, and incubated at 95°C for 10 min. Proteins were then resolved on a SDS–PAGE gel at 150 V and transferred to a nitrocellulose membrane by Western blotting at 300 mA. Membrane was blocked with 2% milk in PBST and stained with anti-FUS antibody (HPA008784, Sigma-Aldrich) diluted 1:10,000 in 2% milk in PBST. The membrane was then stained with HRP-conjugated secondary antibody (Sigma-Aldrich) diluted 1:5,000 in 2% milk in PBST. Membranes were covered with the substrate Luminata Crescendo (Millipore), and chemiluminescence was detected on Amersham Hyperfilm ECL (GE Healthcare).

### RNase microinjection

Cells were grown in 3.5-cm glass bottom dishes (MatTek) and exposed to 2-h heat stress. Afterward, the growth medium was supplemented with 20 mM HEPES, and cells were maintained at 37°C and imaged on a spinning-disk confocal microscope (Olympus IX81) using 60× oil immersion objective (UPlanSA; 1.35NA). 50 ng RNase A (Carl Roth) diluted in 1 nl PBS was injected into cells using FemtoJet microinjector (Eppendorf). Images were taken before and after RNase injection. Fiji was used for SG segmentation (based on G3BP1-mCherry signal) and subsequent quantification of SG area. The reduction of SG area upon RNase injection was plotted in MS Excel.

### Fluorescence *in situ* hybridization

Cells expressing SOD1(A4V)-GFP were exposed to heat stress for 2 h and fixed with 4% PFA (diluted in PBS) for 10 min. This was followed by washing with PBS and then permeabilization with 0.2% Triton X-100 (diluted in PBS) for 10 min. The cells were then washed with 2× SSC. Hybridization was then performed by incubating the cells for 2 h at 37°C in 4× SSC containing 10% formamide, 5% dextran sulfate, 1% BSA, 0.5 mM EDTA, and 100 nM biotinylated oligo-dT probe (23-mer). This was followed with washing in 2× SSC (3 × 10 min) and blocking in 3% BSA (diluted in 4× SSC) for 1 h at room temperature. The cells were then incubated with primary antibodies (goat anti-biotin, Sigma-Aldrich, B3640; rabbit anti-G3BP1, Thermo Fisher Scientific, PA5-29455) diluted in 1% BSA (in 4× SSC) for 1 h. The cells were then washed in 4× SSC and incubated with secondary antibodies conjugated with Alexa Fluor fluorophores (Invitrogen) diluted in 1% BSA (in 4× SSC) for 1 h. The cells were then washed in 4× SSC, then washed in 2× SSC, and finally mounted in DAPI-Fluoromount G (SouthernBiotech). The cells were then imaged using the DeltaVision imaging system (Applied Precision) as described above.

**Expanded View** for this article is available online.

## Acknowledgements

We thank Ina Poser, Doris Richter, and Andrej Pozniakovski for help with generating expression constructs and human cell lines. We thank Bert Nitzsche, Britta Schroth-Diez, and Jan Peychl (Light Microscopy Facility) for expert assistance with imaging; Benoit Lombardot for assistance with image analysis; Barbara Borgonovo (Chromatography Facility); and David Drechsel and Regis Lemaitre (Protein Expression and Purification Facility) for help with protein expression and purification. We gratefully acknowledge funding from the Max Planck Society, the Alexander von Humboldt Foundation (GRO/1156614 STP-2 to AP), and the German Federal Ministry of Research and Education (BMBF 031A359A MaxSynBio) to TMF. DM was supported by a grant of the DFG Center for Regenerative Therapies (CRTD) in Dresden. This is an EU Joint Programme—Neurodegenerative Disease Research (JPND) project. The project is supported through the following funding organizations under the aegis of JPND—www.jpnd.eu (list of national/regional organizations who are funding project, by country, in alphabetical order), for example, France, Agence National de la Recherche; United Kingdom, Medical Research Council. This project has received funding from the European Union's Horizon 2020 research and innovation program under grant agreement no. 643417.

## Author contributions

DM performed all cell culture experiments and performed all image analysis. TMF, AK, and EEB purified and characterized wild-type and mutant Ubc9. AP performed the FUS *in vitro* aging assay with the help of TMF. HOL constructed BAC cell lines and shared ideas. SM performed RNase microinjection. SC and AAH shared important insights and ideas. AAH provided important resources. SA conceived the project, and SA and DM wrote the manuscript.

## Conflict of interest

The authors declare that they have no conflict of interest.

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
