## [Review Process File · The EMBO Journal]

Manuscript EMBO-2016-95957

An aberrant phase transition of stress granules triggered by misfolded protein and prevented by chaperone function

Daniel Mateju, Titus M. Franzmann, Avinash Patel, Andrii Kopach, Edgar E. Boczek, Shovamayee Maharana, Hyun O. Lee, Serena Carra, Anthony A. Hyman and Simon Alberti

Corresponding author: Simon Alberti, Max Planck Institute of Molecular Cell Biology and Genetics

Review timeline:

Submission date:	25 October 2016
Editorial Decision:	04 January 2017
Revision received:	13 February 2017
Editorial Decision:	07 March 2017
Accepted:	07 March 2017

Editor: Anne Nielsen

Transaction Report:

1st Editorial Decision

04 January 2017

Thank you for submitting your manuscript for consideration by The EMBO Journal and my apologies for the extended duration of the review period. Your manuscript has now been seen by three referees whose comments are shown below.

As you will see from the reports, referees #2 and #3 express high interest in the findings reported in your manuscript and recommend publication, following the inclusion of a number of additional control experiments and clarifications. At the same time referee #1 is concerned about the partial overlap with previously published work from both your lab and others, and consequently asks for a more extensive characterization of the difference between normal and aberrant stress granules.

Given the referees' overall positive recommendations, I would like to invite you to submit a revised version of the manuscript, addressing the comments of all three reviewers. In our view, the constructive suggestions and requests from refs #2 and #3 would also go a long way to address the major concern raised by ref #1. I should add that it is EMBO Journal policy to allow only a single round of revision, and acceptance or rejection of your manuscript will therefore depend on the completeness of your responses in this revised version.

For the revised manuscript I would particularly ask you to focus your efforts on the following points:

- > Clarify all points related to experimental strategy, sample acquisition and quantification.
- > Expand data on the selective accumulation of mutant SOD1 in SGs as pointed out by refs #1 and

#2

-> In our view the suggestions from ref #2 to use Super resolution microscopy for improved insights on the structure and dynamics of the SGs would clearly add to the manuscript; however, we also realize that this may be technically difficult to establish during revision.

REFEREE REPORTS

Referee #1:

The manuscript by Mateju et al investigates the interaction between stress granule- associated proteins and misfolded proteins, mainly disease-linked mutants of SOD1. Stress granule resident proteins are hot spots for ALS-causing mutations, as is the SOD1 protein. The authors set out to investigate the mechanism of converting physiological SGs into pathological entities. The groups that collaborated in the study have authored a number of important papers on this topic in the past. This work is a continuation of a recent paper in Molecular Cell, where the authors propose that a chaperone network mediates SG disassembly.

The proposed novelty of this manuscript rests almost entirely on the interpretation of colocalization dynamics of SG proteins and SOD mutants. Otherwise, many of the observations are somewhat similar to previous reports (for example Farrarwell et al., 2015 - showing that mutant and wt FUS and TDP-43 partition to different kinds of inclusions, some of which can recruit wt SG components. Also, the idea that SOD mutants can co-aggregate with less aggregation-prone proteins and substantially decrease their mobility by titrating Hsp70 and other proteostasis components is also well established.

In the view of this reviewer the novelty of this paper hinges on proving that there is a mechanistic distinction between what the authors call normal and aberrant SGs. SGs are an under-defined entity, with potentially aggregation-prone and (albeit slightly) overexpressed proteins being used to mark them. Without a more extensive characterization of normal versus aberrant SGs it is very difficult to be convinced that there is a distinction between SOD mutants "infecting" normal SGs and the opposite scenario, namely SOD mutants aggregating and inducing the co-aggregation of metastable SG proteins.

Major points:

1. Figure EV2D does not show the SGs and aggresome precursors simultaneously. The data only argue that ubc aggregates G3BP is already out of visible puncta.
2. The authors argue that in their system SG proteins are expressed at physiological levels. However, it's not endogenous proteins that are being tagged here - there is at least a 2-fold over-expression. For metastable aggregation prone proteins this may influence their aggregation kinetics.
3. Page 7 line 22 is not clear at all. How is the data showing that "recruitment to SGs is specific for misfolded variants of SOD1" when all SOD variants accumulate?
4. Page 8, line 9 - Fluorescence microscopy does not provide enough information to conclude that SGs are completely devoid of mutant SOD.

Referee #2:

Mateju and colleagues present an interesting manuscript that contains experimentally robust and thoughtful data using a number of technologically impressive techniques. They describe a novel mechanism by which misfolded proteins are enriched in cytoplasmic compartments known as a stress granules (SGs) prior to clearance by two independent mechanisms, involving chaperones and autophagy. This work is consistent with a number of trending subjects within the fields of cellular bodies and stress responses (concentration-dependent phase transitions and autophagy, respectively.) The experiments are well described and logical, which is emphasized by the well-written manuscript. I have some small comments and suggestions to improve the manuscript, but strongly believe that this is suitable for publication in the EMBO Journal.

Figure 1:

- 1) Do Ubc9s or SOD1 mutants form SG-like structures in vitro in the absence of FUS or other in vitro nucleators of body formation (such as TDP-43, which the authors identified as decreased in SG localization following prolonged heat shock)?
- 2) Regarding the high throughput experiments, how did the authors decide upon the signal intensity threshold, and what was it?
- 3) In later figures, the authors state that there was a mostly binary accumulation of misfolded proteins in SGs.
- 4) Please provide cell-level % in addition to SG-level analysis.
- 5) Did the authors observe a change in SG number under different conditions? This is especially relevant for later figures, where a change in protein composition was reported over time. Where these changes in total SG composition reflective of protein relocalization/degradation or the formation of new SGs?
- 6) Throughout all high throughput imaging experiments, can the authors correlate protein expression with SOD1+ SG appearance (or other factors, such as SG size or cell cycle)?
- 7) There is a typo on page 7, line 21 - Figure 2G should be Figure 1G.
- 8) The authors may want to consider using another method to induce protein misfolding or SG assembly (such as ...) to show that this is a universal phenomenon. Do misfolded SOD1 proteins accumulate naturally in SGs in laminopathy cell models, as a model system to study ageing?
Figure 2
- 9) The authors state that mutant SOD1 is aggregated and accumulated in SGs but provide no direct evidence for this.
- 10) 3D super resolution microscopy (rather than the 2D projected images shown) that are modeled using software such as Imaris (Bitplane) may indicate whether misfolded protein localization to the SG is a shell or more dispersed throughout the whole structure.
- 11) The manuscript will be vastly improved by a temporal dissection of SG ultrastructure using structural illumination microscopy (SIM). Is there a change in misfolded protein localization that correlates with the suggested decrease in RBP SG localization?
Figure 3
- 12) Please provide mean square displacement (MSD, μm^2) data to support changes in SG movement.
Figure 4
- 13) Did the authors observe any changes in SG size or number with composition?
- 14) 4C - please try to standardize the axes for reader comprehension.
- 15) EV4B - Were these changes due to protein degradation, relocalization away from the SG, or the creation of new SGs at these time points? Thus, was the number of DRiP+ SGs stable, but more SGs were assembled that don't contain DRiPS?
Figure 5
- 16) Does HSP70 regulate accumulation of misfolded proteins in SGs in vitro?
Figure 6
- 17) Do other PI3K inhibitors (LY294002 etc) also increase SG-aggresome association?

Referee #3:

In this paper the authors investigate how unstable and aggregation prone proteins (mutants of Ubc and SOD1) interact with stress granules (bodies positive for FUS or G3BPs) and change their physical properties. They find that stress granules containing additional protein aggregates mature over time, and that the aggregated proteins somehow displace the initial SG components, leading to less dynamic and more solid-like cellular inclusions. The authors then go on to investigate the impact of heat-shock proteins 70 and 27 on SGs, and find that HSP presence correlates with that of aggregated proteins. Interestingly, an inhibitor of hsp70 manipulates the aggregates in a predictable way. Finally, the authors link hard-to-clear SGs with the aggresome, suggesting that cells possess multiple pathways for dealing with stress, and clearing its causalities.

Overall the paper presents a very interesting set of experiments, analysing the relationship between aggregating proteins, chaperones and stress granules. This reviewer recommends this paper for publication, with the following minor points for consideration.

Minor points

1. Figure 1A/B

"Interestingly, misfolded Ubc9TS accumulated in FUS compartments while Ubc9WT remained largely in solution (Fig 1A). This suggests that misfolded proteins may have a tendency to accumulate in phase-separated liquid compartments."

From the images in the figure, when I manually adjust the contrast, it looks like there is zero Ubc9WT present in \emptyset or WT panels, either in solution or in FUS compartments. It's not clear from these images that the protein has remained in solution. Perhaps it would be clearest if the change in partitioning were quantified as they do later in the paper? The change in partitioning looks like it's rather large.

2. "9% of SGs were highly enriched for Ubc9TS".

Presumably this is because of heterogeneity in the SGs rather than mass action? I think it would be interesting if this point were discussed further.

Is this (and other comparisons coming later in the manuscript) at the per-cell level or over a large population of cells? Is the high-content imaging data normalised for protein expression level? Also, what are the properties of this population of SGs - size, shape, number distribution compared to the remaining 91%? Presumably this data is accessible from the high-content automated imaging assay? In short, is it obvious why some of the SGs are enriched, but others are not?

3. What are the errors associated with the intensity ratios in Fig 4B?

4. Related to Fig 5B.

How do the authors explain the apparent depletion of ubiquitin from the nucleus in the Ars+DMSO condition? Generally, the ubiquitin staining in Fig 5B appears mostly not localised to SGs. Is there a high degree of non-specific staining here, or something else going on? The ubiquitin staining in Fig EV6, by contrast, looks by eye far more localised to SGs.

5. In the introduction, the authors note that SGs contain a significant amount of RNA and are involved in translational repression during stress. Is there any indication of the RNA content of aberrant SGs is different to that of normal SGs?

1st Revision - authors' response

13 February 2017

Point-by-point reply to reviewers

Reviewer 1: *The manuscript by Mateju et al investigates the interaction between stress granule-associated proteins and misfolded proteins, mainly disease-linked mutants of SOD1. Stress granule resident proteins are hot spots for ALS-causing mutations, as is the SOD1 protein. The authors set out to investigate the mechanism of converting physiological SGs into pathological entities. The groups that collaborated in the study have authored a number of important papers on this topic in the past. This work is a continuation of a recent paper in Molecular Cell, where the authors propose that a chaperone network mediates SG disassembly.*

Reply: We thank the reviewer for the feedback.

Reviewer 1: *The proposed novelty of this manuscript rests almost entirely on the interpretation of colocalization dynamics of SG proteins and SOD mutants. Otherwise, many of the observations are somewhat similar to previous reports (for example Farrarwell et al., 2015 - showing that mutant and wt FUS and TDP-43 partition to different kinds of inclusions, some of which can recruit wt SG components. Also, the idea that SOD mutants can co-aggregate with less aggregation-prone proteins and substantially decrease their mobility by titrating Hsp70 and other proteostasis components is also well established.*

In the view of this reviewer the novelty of this paper hinges on proving that there is a mechanistic distinction between what the authors call normal and aberrant SGs. SGs are an under-defined

entity, with potentially aggregation-prone and (albeit slightly) overexpressed proteins being used to mark them. Without a more extensive characterization of normal versus aberrant SGs it is very difficult to be convinced that there is a distinction between SOD mutants "infecting" normal SGs and the opposite scenario, namely SOD mutants aggregating and inducing the co-aggregation of metastable SG proteins.

Reply: We agree with the reviewer that SGs are an underdefined entity. For this reason, we are very careful to make sure that we study SGs and no other aggregate or assembly. This is also why we use multiple markers for SGs, such as FUS, G3BP, eIF3 (Figs 1E, 1F, Appendix Fig S2A). In some experiments, we had to rely on GFP-tagged SG proteins for live cell imaging, but in others we used immunofluorescence using cells that do not overexpress SG markers (Figs 4A, Appendix Fig S2A). Our data are very clear: aberrant SGs that contain misfolded proteins also contain all the SG markers mentioned above (Figs 1E, 1F, Appendix Fig S2A). To address the reviewer's concern further, we performed fluorescence in situ hybridization, which revealed that both normal SGs and aberrant SGs contain poly(A) mRNA in similar amounts (Fig EV3A). In this specific experiment, no SG marker was overexpressed and SGs were detected by immunofluorescence. Thus, we conclude that aberrant SGs are indeed SGs: they contain several key SG marker proteins and have a high content of poly(A) mRNA.

The reviewer suggests that the opposite scenario could also be true, namely that SOD1 aggregates sequester and induce the aggregation of SG proteins. This was also one of the concerns that we had initially, but we could rule this out by performing a time-resolved analysis of SG formation and maturation. In time-lapse experiments with live cells, we observed SG formation before SOD1 formed visible aggregates (Figs 2A, 2B). In addition, we could directly observe the conversion of physiological SGs into aberrant SGs in some time-lapse movies (Movie EV5). Moreover, using FRAP, we find that SOD1 becomes immobilized only after prolonged heat stress, when SGs are already present (Figs 2C, 2D). Based on these findings, we conclude that SOD1 is recruited to SGs and not vice versa.

We agree with the reviewer that it is important to show a clear difference between normal and aberrant SGs. In order to provide such evidence, we investigated whether the structural integrity of aberrant SGs depends on RNA. The rationale was the following: physiological SGs are dynamic membrane-less compartments, whose formation depends on free mRNA (Kedersha *et al*, 2000; Bounedjah *et al*, 2014). However, aberrant SGs increasingly behave as protein aggregates with reduced dynamics, suggesting that they are no longer dependent on RNA. To test this hypothesis, we microinjected RNase A into heat-stressed cells that contained visible SGs. As expected, RNase injection triggered the rapid disintegration of SOD1-negative SGs. Remarkably, however, the same treatment had no visible effect on SOD1-positive SGs (Figs 3G, H). Based on these data, we conclude that aberrant SGs are protein-based assemblies, whereas physiological SGs are RNA-based assemblies. This suggests that the conversion into aberrant SGs involves a gradual replacement of physiological RNA-protein interactions with aberrant protein-protein interactions. It is easy to understand that this affects the dynamics of SGs. It is also clear that this creates a strong dependence on chaperones for disassembly, as observed in Figure 5C and D. We think that this experiment adds important weight to our study. We have therefore added these data to Figure 3.

Reviewer 1: Major points: 1. Figure EV2D does not show the SGs and aggresome precursors simultaneously. The data only argue that *ubc* aggregates G3BP is already out of visible puncta.

Reply: We agree. The simultaneous presence of SGs and aggresome precursors was visible only in a short time window in this experiment (between 4 and 6 h of MG132 treatment). For this reason, we added a new time point (Fig EV2D is now Fig EV1D). The new panel (5 h) shows a cell in which both SGs and aggresome precursors are visible but are not colocalizing. This can also be seen in Movie EV1.

Reviewer 1: 2. The authors argue that in their system SG proteins are expressed at physiological levels. However, it's not endogenous proteins that are being tagged here - there is at least a 2-fold over-expression. For metastable aggregation prone proteins this may influence their aggregation kinetics.

Reply: Because BAC transgenes have all the endogenous promoter and enhancer elements, they autoregulate their expression levels, and the amount of protein produced usually only deviates slightly from normal levels, if at all. To estimate the expression level of the BAC transgenes, we included an immunoblot. Based on these data, the expression level does not seem to be higher than 2-fold overexpression (Fig EV1A). To make sure that this slight increase in expression levels does not affect the aggregation propensity of the proteins, we generated different cell lines in which we tagged FUS or G3BP with different fluorophores (GFP or mCherry). The colocalization with misfolded proteins was not different between these cell lines (Figs 1B, C, E, F). We also performed experiments, where we detected endogenous SG markers using immunofluorescence. For these cells, we observed a similar number of aberrant SGs as in the presence of BAC transgenes. We therefore conclude that the presence of the transgenes does not affect protein aggregation.

Reviewer 1: 3. Page 7 line 22 is not clear at all. How is the data showing that "recruitment to SGs is specific for misfolded variants of SOD1" when all SOD variants accumulate?

Reply: We consistently saw that mutant SOD1 variants were much more strongly enriched in SGs than wild type SOD1 (Fig 1G). All the mutant SOD1 variants that we tested (A4V, G37R, G85R, G93A) have been shown to be prone to misfolding and aggregation, in contrast to wild type SOD1 (Rakhit *et al*, 2007; Prudencio *et al*, 2009). We made changes to the text to make this more clear.

Reviewer 1: 4. Page 8, line 9 - Fluorescence microscopy does not provide enough information to conclude that SGs are completely devoid of mutant SOD.

Reply: We agree that "devoid" is an inaccurate term. We have changed the text to address this.

Reviewer 2: Mateju and colleagues present an interesting manuscript that contains experimentally robust and thoughtful data using a number of technologically impressive techniques. They describe a novel mechanism by which misfolded proteins are enriched in cytoplasmic compartments known as a stress granules (SGs) prior to clearance by two independent mechanisms, involving chaperones and autophagy. This work is consistent with a number of trending subjects within the fields of cellular bodies and stress responses (concentration-dependent phase transitions and autophagy, respectively.) The experiments are well described and logical, which is emphasized by the well-written manuscript. I have some small comments and suggestions to improve the manuscript, but strongly believe that this is suitable for publication in the EMBO Journal.

Reply: We thank the reviewer for the positive feedback.

Reviewer 2: Figure 1: 1) Do Ubc9s or SOD1 mutants form SG-like structures in vitro in the absence of FUS or other in vitro nucleators of body formation (such as TDP-43, which the authors identified as decreased in SG localization following prolonged heat shock)?

In the absence of FUS, purified Ubc9TS remains in solution and does not form droplets. We now include this data in Appendix Figure S1D.

Reviewer 2: 2) Regarding the high throughput experiments, how did the authors decide upon the signal intensity threshold, and what was it?

Reply: We chose an intensity ratio above which the accumulation of proteins in SGs is clearly visible by eye. As specified in the figure legends, we used a threshold of 1.4 for Ubc9 and SOD1.

Reviewer 2: 3) In later figures, the authors state that there was a mostly binary accumulation of misfolded proteins in SGs.

Reply: In reality, the accumulation of misfolded proteins in SGs was not binary, but there were gradual differences in the amount of accumulated misfolded protein in SGs. However, in order to see clear functional differences, we focussed on the extreme cases, rather than on SGs with an intermediate accumulation of SOD1. This was necessary because SGs with intermediate levels of SOD1 showed large variability in their properties and dynamics. We made changes to the text to make this more clear.

Reviewer 2: 4) Please provide cell-level % in addition to SG-level analysis.

Reply: We now provide these data in the Appendix Figure S2C, D. This new analysis shows the fraction of cells (in percent) containing aberrant SGs enriched for SOD1(A4V) or Ubc9TS. For cells that expressed misfolding-prone instead of wild type control proteins, there was a much higher probability that they contained one or more aberrant SGs per cell.

Reviewer 2: 5) Did the authors observe a change in SG number under different conditions? This is especially relevant for later figures, where a change in protein composition was reported over time. Where these changes in total SG composition reflective of protein relocalization/degradation or the formation of new SGs?

Reply: We do not think that the changes at later time points occur because of the formation of new SGs, but rather reflect the redistribution of proteins from/to SGs. In our time-lapse imaging experiments, SGs only formed in the first 60 minutes after stress onset. During prolonged stress (60-150 minutes), formation of new SGs was a very rare event. To test this further, we quantified the number of SGs per cell over time. This revealed that the SG number per cell was slightly decreasing after 60 minutes of heat stress, which is probably due to SG fusion (see Figure R1 below). Therefore, we can exclude that the formation of new SGs during prolonged stress affects SG composition.

Figure R1: SG number per cell during heat stress. The number of SGs was counted based on eIF3 η immunofluorescence. SG number per cell is plotted for different time points of heat stress. Mean values from 3 experiments are shown. At least 100 cells were counted for each sample (1 replicate of 1 time point). Error bars = SEM.

Reviewer 2: 6) Throughout all high throughput imaging experiments, can the authors correlate protein expression with SOD1+ SG appearance (or other factors, such as SG size or cell cycle)?

Reply: We did not see any obvious correlations with protein expression or cell cycle, but this data cannot easily be extracted from our high-content imaging data. To gain more insight, we analyzed SG size and circularity (Appendix Figure S5). There is no obvious association between SG circularity and enrichment of SOD1(A4V) or Ubc9TS in SGs. Interestingly, however, we find that SOD1-positive SGs (and Ubc9TS-positive SGs) were on average slightly larger, but this difference was not statistically significant. We speculate that this could be due to the fact that aberrant SGs are less dynamic and thus become larger with time relative to physiological SGs.

Reviewer 2: 7) There is a typo on page 7, line 21 - Figure 2G should be Figure 1G.

Reply: Thank you, we changed this.

Reviewer 2: 8) The authors may want to consider using another method to induce protein misfolding or SG assembly (such as) to show that this is a universal phenomenon. Do misfolded

SOD1 proteins accumulate naturally in SGs in laminopathy cell models, as a model system to study ageing?

Reply: This is a very good idea and something that we will definitely test in the future. However, we feel that for this study, these experiments are beyond the scope. We would like to mention though that an increasing number of aberrant SGs have very recently been reported for aging *C. elegans* (Lechler *et al.*, 2017), suggesting that there may be an interesting connection between SG homeostasis and aging.

Reviewer 2: *Figure 2, 9) The authors state that mutant SOD1 is aggregated and accumulated in SGs but provide no direct evidence for this.*

Reply: Figures 2A and 2B show the accumulation of mutant SOD1 in SGs. The evidence for aggregation comes from the FRAP experiments in Figures 2C-G and from super-resolution microscopy in Figures 2H, 2I. In the revised manuscript, we now provide additional data showing that aberrant SGs have properties of protein aggregates (Fig 3G). We find that aberrant SGs become independent of RNA, as they remain intact after degradation of RNA by RNase A microinjection. This is consistent with the idea that SGs convert from an RNA-based assembly into protein-based aggregates.

Reviewer 2: *10) 3D super resolution microscopy (rather than the 2D projected images shown) that are modeled using software such as Imaris (Bitplane) may indicate whether misfolded protein localization to the SG is a shell or more dispersed throughout the whole structure.*

Reply: We now provide 3D visualisation of the image stacks obtained by structured illumination microscopy (Movie EV2). The mutant SOD1 indeed seems to be localized mainly (but not exclusively) at the surface of the SGs.

Reviewer 2: *11) The manuscript will be vastly improved by a temporal dissection of SG ultrastructure using structural illumination microscopy (SIM). Is there a change in misfolded protein localization that correlates with the suggested decrease in RBP SG localization?*

Reply: We performed SIM at two different timepoints of heat stress (EV2C, D). The obtained images revealed that distinct regions of SOD1(A4V) accumulation within SGs appear only with prolonged stress. However, it is difficult to say from these data whether the formation of these regions is responsible for the partial depletion of RBPs from SGs.

Reviewer 2: *Figure 3, 12) Please provide mean square displacement (MSD, μm^2) data to support changes in SG movement.*

Reply: In the cells that contain two populations of SGs, we noticed that SOD1-positive SGs were more mobile than SOD1-negative SGs (Movie EV3). However, there generally seems to be a large variability in SG movement, which might depend on stress duration and other factors. We therefore think that further analysis of SG mobility would be very difficult. For this reason, we have made changes to the text, and we no longer make claims about the differences in SG mobility between SOD1-positive and SOD1-negative SGs.

Reviewer 2: *Figure 4, 13) Did the authors observe any changes in SG size or number with composition?*

Reply: We see very little correlation with SG size or circularity (Figure R2). The SG number per cell changes over time during heat stress (Figure R1), but this does not explain the observed changes of SG composition with time (Figure 4D).

Figure R2. Correlation between SG composition and SG size or circularity. Using the data shown in Figure 4C and Appendix Figure S6, SG area and circularity was measured and correlated with the enrichment of HSP27, HSP70 or TDP-43 in the SG. 1000 SGs are plotted in each graph. Pearson's correlation coefficient (r) is shown.

Reviewer 2: 14) 4C - please try to standardize the axes for reader comprehension.

Reply: This has been addressed. Furthermore, we added data for ubiquitin to Figure 4C. The data for HSP70 is now in Appendix Figure S6.

Reviewer 2: 15) EV4B - Were these changes due to protein degradation, relocalization away from the SG, or the creation of new SGs at these time points? Thus, was the number of DRiP+ SGs stable, but more SGs were assembled that don't contain DRiPS?

Reply: Degradation and relocalization are both possible. As explained above (point 5), creation of new SGs is unlikely to play a major role during later stages of heat stress. In this experiment, SG number per cell was decreasing after 60 minutes of stress, presumably because of SG fusion (Figure R1). Therefore, these changes in SG composition likely reflect protein relocalization or degradation, rather than formation of new SGs.

Reviewer 2: Figure 5, 16) Does HSP70 regulate accumulation of misfolded proteins in SGs in vitro?

Reply: This is a good suggestion. Indeed, our goal in the future is to in vitro reconstitute the protein quality control machinery of RNP granules. However, this will require a lot of additional time and effort. We therefore feel that this is beyond the scope of this study.

Reviewer 2: Figure 6, 17) Do other PI3K inhibitors (LY294002 etc) also increase SG-aggresome association?

Reply: We repeated the experiment using LY294002. Similar to treatment with wortmannin, we observed increased localization of SGs at the aggresome in LY294002-treated cells (Figure R3). However, we found that the LY294002-treated cells in this experiment also contained aggresomes more frequently than control-treated cells, which makes the interpretation of this experiment more complicated than the one with wortmannin.

Figure R3. Effect of LY294002 on SG-aggresome association. Cells expressing G3BP2-GFP and Ubc9TS-mCherry were treated with 50 μ M LY294002 (Cell Signaling) or DMSO during heat stress (2 h) and subsequent recovery (6 h). After fixation, at least 100 cells expressing Ubc9TS were examined to determine the localization of SGs at the aggresome region. Mean values from 3 experiments are shown. Error bars = SEM. * $p < 0.05$ (t-test).

Reviewer 3: *In this paper the authors investigate how unstable and aggregation prone proteins (mutants of Ubc and SOD1) interact with stress granules (bodies positive for FUS or G3BPs) and change their physical properties. They find that stress granules containing additional protein aggregates mature over time, and that the aggregated proteins somehow displace the initial SG components, leading to less dynamic and more solid-like cellular inclusions. The authors then go on to investigate the impact of heat-shock proteins 70 and 27 on SGs, and find that HSP presence correlates with that of aggregated proteins. Interestingly, an inhibitor of hsp70 manipulates the aggregates in a predictable way. Finally, the authors link hard-to-clear SGs with the aggresome, suggesting that cells possess multiple pathways for dealing with stress, and clearing its causalities.*

Overall the paper presents a very interesting set of experiments, analysing the relationship between aggregating proteins, chaperones and stress granules. This reviewer recommends this paper for publication, with the following minor points for consideration.

Reply: We thank the reviewer for the positive feedback.

Reviewer 3:

Minor points

1. Figure 1A/B "Interestingly, misfolded Ubc9TS accumulated in FUS compartments while Ubc9WT remained largely in solution (Fig 1A). This suggests that misfolded proteins may have a tendency to accumulate in phase-separated liquid compartments." From the images in the figure, when I manually adjust the contrast, it looks like there is zero Ubc9WT present in \square or WT panels, either in solution or in FUS compartments. It's not clear from these images that the protein has remained in solution. Perhaps it would be clearest if the change in partitioning were quantified as they do later in the paper? The change in partitioning looks like it's rather large.

Reply: We added a quantification of Ubc9 accumulation in FUS droplets (Appendix Figure S1C). We also changed the contrast settings for the images, but the Ubc9WT signal is still much less bright than Ubc9TS. The reason for this is that Ubc9WT is diluted in solution and not enriched in foci as Ubc9TS.

Reviewer 3: *2. "9% of SGs were highly enriched for Ubc9TS". Presumably this is because of heterogeneity in the SGs rather than mass action? I think it would be interesting if this point were discussed further. Is this (and other comparisons coming later in the manuscript) at the per-cell level or over a large population of cells? Is the high-content imaging data normalised for protein expression level? Also, what are the properties of this population of SGs - size, shape, number distribution compared to the remaining 91%? Presumably this data is accessible from the high-content automated imaging assay? In short, is it obvious why some of the SGs are enriched, but others are not?*

Reply: Indeed, enrichment of misfolded proteins in SGs is due to the heterogeneity of SGs, largely between cells, which may be due to the health or age of these cells. However, we not only observed variability between cells, but also identified cells that contained both physiological and aberrant SGs at the same time, which argues against a simple mass action mechanism. In addition, aberrant SGs also formed in the absence of overexpressed proteins (Fig 5B), further suggesting that aberrant SGs are not an artifact of protein overexpression. Generally, a correlation with expression levels was not apparent from our data.

Our high-content imaging assay analyzes a large number of individual SGs over a large population of cells. In our assay, the fluorescence intensity is normalized to the area around SG, which reflects cytosolic levels. In the revised manuscript, we have added additional cell-level analyses (Appendix Figs S2C, D). To gain more insight, we also analysed SG size, shape and number (Appendix Figure S5). SG circularity did not correlate with the enrichment of SOD1(A4V) or Ubc9TS in SGs. Interestingly, SGs enriched for SOD1(A4V) or Ubc9TS were on average slightly larger and they were more prevalent in cells with a lower number of SGs. However, these were relatively small differences that cannot explain the observed variability in SG composition and other observations, such as decreased G3BP1 turnover (Fig 3E), reduced dynamics (Fig 3D) or RNase insensitivity (Fig 3H).

Reviewer 3: 3. *What are the errors associated with the intensity ratios in Fig 4B?*

Reply: The enrichment ratios were calculated from the mean fluorescence intensities within the indicated ROIs, which were automatically segmented based on the signal of the SG marker. The whole distribution of the ratios can be seen in Figure 4C. We verified that our assay does not detect other structures than SGs. However, not all SGs in a sample could be detected, due to occasional autofocus errors or highly irregular (fuzzy) edges of some SGs. Some errors might be created due to SGs being closely adjoined. In general, the assay was robust and the high enrichment ratios reflected an accumulation that was visible by eye.

Reviewer 3: 4. *Related to Fig 5B. How do the authors explain the apparent depletion of ubiquitin from the nucleus in the Ars+DMSO condition? Generally, the ubiquitin staining in Fig 5B appears mostly not localised to SGs. Is there a high degree of non-specific staining here, or something else going on? The ubiquitin staining in Fig EV6, by contrast, looks by eye far more localised to SGs.*

Reply: In Figure 5B, we do not use heat stress and we compare four different conditions. The Ars+DMSO condition is the only conditions in which proteostasis is unimpaired and ubiquitinated proteins can be efficiently degraded. The other three conditions all involve some form of proteostasis inhibition (+ VER or + MG132 or both). We only observe a strong depletion of ubiquitin signal in the nucleus in the Ars+DMSO condition with no proteostasis impairment. Instead, when proteostasis is impaired (+VER or +MG132), ubiquitin is enriched in the nucleus. The strongest ubiquitin enrichment is seen, when both MG132 and VER are added to cells. This is in agreement with recent findings that ubiquitylated proteins are degraded by nuclear proteasomes (Park *et al*, 2013), and suggests that upon inhibition of the proteostasis machinery substrates of the ubiquitin/proteasome system accumulate in the nucleus. We have added a sentence to the figure legend to explain the difference in ubiquitin staining.

In Figure 5B, the accumulation of ubiquitin in SGs is only visible in the condition where VER is present. The other two conditions do not show a significant enrichment of ubiquitin. The different distribution of ubiquitin compared to Figure EV5B (previously EV6) is because of the different conditions used. In Figure EV5B we used heat stress, which in our experience causes more protein misfolding and thus stronger recruitment of ubiquitin to SGs. Additionally, the co-expression of Ubc9TS might have further enhanced this effect. Thus, we do not think that there is a problem with the specificity of the antibody, but that the differences in staining can be explained by the different conditions used.

Reviewer 3: 5. *In the introduction, the authors note that SGs contain a significant amount of RNA and are involved in translational repression during stress. Is there any indication of the RNA content of aberrant SGs is different to that of normal SGs?*

Reply: To address this concern, we performed fluorescence in situ hybridization on heat-stressed cells expressing SOD1(A4V)-GFP. We observed a strong poly(A) RNA signal in both SOD1-negative and SOD1-positive SGs, without any noticeable difference in the RNA content. This suggests that aberrant SGs still contain mRNA. In the revised version of the manuscript, we also include new data showing that the structural integrity of aberrant SGs does not depend on RNA. To investigate the role of RNA in SG assembly, we microinjected RNase A into heat-stressed cells to degrade RNA. RNA degradation dissolved SOD1- negative SGs, but had minimal effect on SOD1-positive SGs (Figs 3G, H). Therefore, RNA is still a component of aberrant SGs, but does not seem to be required for the structural integrity of these SGs. We thank the reviewer for asking us to perform these additional experiments, which we think have significantly improved the paper.

References

- Bounedjah O, Desforgues B, Wu T-D, Pioche-Durieu C, Marco S, Hamon L, Curmi PA, Guerquin-Kern J-L, Piétremont O & Pastré D (2014) Free mRNA in excess upon polysome dissociation is a scaffold for protein multimerization to form stress granules. *Nucleic Acids Res.* **42**: 8678–8691
- Kedersha N, Cho MR, Li W, Yacono PW, Chen S, Gilks N, Golan DE & Anderson P (2000) Dynamic shuttling of TIA-1 accompanies the recruitment of mRNA to mammalian stress granules. *J. Cell Biol.* **151**: 1257–1268
- Lechler MC, Crawford ED, Groh N, Widmaier K, Jung R, Kirstein J, Trinidad JC, Burlingame AL & David DC (2017) Reduced Insulin/IGF-1 Signaling Restores the Dynamic Properties of Key Stress Granule Proteins during Aging. *Cell Rep.* **18**: 454–467
- Park S-H, Kukushkin Y, Gupta R, Chen T, Konagai A, Hipp MS, Hayer-Hartl M & Hartl FU (2013) PolyQ proteins interfere with nuclear degradation of cytosolic proteins by sequestering the Sis1p chaperone. *Cell* **154**: 134–145
- Prudencio M, Hart PJ, Borchelt DR & Andersen PM (2009) Variation in aggregation propensities among ALS-associated variants of SOD1: correlation to human disease. *Hum. Mol. Genet.* **18**: 3217–3226
- Rakhit R, Robertson J, Velde CV, Horne P, Ruth DM, Griffin J, Cleveland DW, Cashman NR & Chakrabartty A (2007) An immunological epitope selective for pathological monomer-misfolded SOD1 in ALS. *Nat. Med.* **13**: 754–759

2nd Editorial Decision

07 March 2017

Thank you for submitting a revised version of your manuscript to The EMBO Journal. It has now been seen by two of the original referees and their comments are shown below. As you will see they both find that all criticisms have been sufficiently addressed and I am therefore happy to let you know that your manuscript has been accepted for publication here.

Corresponding Author Name: Simon Alberti

Manuscript Number: EMBOJ-2016-95957